REGISTERED REPORT PROTOCOL

# Impact of protected areas on deforestation in Madagascar from 2000 to 2023: A pre-analysis plan

Diamondra Ramiandrisoa[1,2,3,4¤a☯]*, Florent Bédécarrats[2,5¤b☯], Melvin H. L. Wong[6☯], Thierry Razanakoto[1,2¤a☯]

**1** Development Centre for Economic Studies and Research (CERED), Economics Department, Faculty of Economics, Management, and Sociology, University of Antananarivo, Antananarivo, Madagascar, **2** Unité Mixte Internationale – Soutenabilité et Résilience (UMI-SOURCE), Guyancourt, France, **3** University of Paris Saclay, Gif-sur-Yvette, France, **4** French National Research Institute for Sustainable Development (IRD), Antananarivo, Madagascar, **5** French National Research Institute for Sustainable Development (IRD), Guyancourt, France, **6** Evaluation Department, KfW Development Bank, Frankfurt am Main, Germany

☯ These authors contributed equally to this work
¤a Current Address: Development Centre for Economic Studies and Research, Economics Department, University of Antananarivo, Antananarivo, Madagascar
¤b Current Address: Unité Mixte Internationale – Soutenabilité et Résilience (UMI-SOURCE), Guyancourt, France
* aina_diamondra.ramiandrisoa@ird.fr

## Abstract

Protected areas are the most prevalent strategy to mitigate biodiversity loss and deforestation, especially in biodiversity hotspots like Madagascar. This pre-analysis plan outlines the data, methods, and identification strategy that will be used to assess the impact on deforestation of terrestrial protected areas created in Madagascar between 2002 and 2022. We will employ coarsened exact matching and difference-in-differences methods to evaluate forest cover loss, leveraging 24 years of high-resolution satellite data. We will incorporate buffer zones to assess spillover and leakage effects. This study addresses limitations of previous ones by combining accurate protected area delimitation and characteristics, a longer temporal coverage, improved characterization of the diverse forest ecosystems, and state-of-the art econometric methods. We will then assess heterogeneity of conservation effects, to better understand the determinants of protected area effectiveness.

## Introduction

Deforestation is a global phenomenon that contributes to the emission of carbon dioxide ($CO_2$) [1]. It is also one of the main drivers of biodiversity loss, undermining the essential ecological functions of ecosystems [2]. This issue is particularly alarming in tropical forests due to their exceptional biodiversity and their critical role in climate regulation [3]. To mitigate deforestation in these regions, many protected areas

**Data availability statement:** All relevant data from this study will be made available upon study completion on an academic repository https://dataverse.ird.fr/.

**Funding:** The authors conducted this study within the framework of the BETSAKA Project. The BETSAKA project is funded by grants from the following institutions. KfW Development Bank: Grant #BMZ 2018-68-538/KfW 108778. The official website of the institution is https://www.kfw.de/About-KfW. French Development Agency (AFD): Grant #CZZ 2557 (PAIRES), available at https://www.afd.fr/fr. French National Research Agency (ANR) : Grant #ANR-22-CPJ1-0052-01. The official website of this institution is https://anr.fr/en/. The support provided by the above-mentioned institutions does not imply their endorsement of the interpretations presented herein, and they bear no responsibility for the content of this publication.

**Competing interests:** The research activities are partially funded by KfW Development Bank in form of a scientific cooperation contract. This contract guarantees that the research activities are conducted independently from any competing interests. The employment of Melvin H. L. Wong at KfW Development Bank and the partial funding of research activities does not alter our adherence to PLOS ONE policies on sharing data and materials.

(PAs) have been established [4]. These areas serve as conservation tools aimed at preserving biodiversity and maintaining the integrity of natural habitats [5].

At the international level, the World Resources Institute (WRI), through its platform reported a loss of 6.7 million hectares of primary tropical forest in 2024. In Madagascar, the loss of primary tropical forest reached 226kha during the same year. This is especially concerning as over 80% of Madagascar's biodiversity is located within forest. This exceptional biodiversity is known as hosting a high degree of endemism and its richness at higher taxonomic level [6] but they are under threat which makes the country a major biodiversity hotspot [7]. Thus, the extinction of any local species would represent a global loss [6].

In addition to its rich biodiversity, Madagascar is among the poorest countries in the world, with 75.2% of the population living below the national poverty line in 2022 [8]. Human activities are the main driver of deforestation in the country [6], particularly agriculture. The remaining forests are threatened by subsistence farming systems inherited from the colonial era [2]. The theory behind the land use change is often based on economic decisions related to opportunity costs [9]. To reduce the deforestation overall and to conserve its unique biodiversity, the Malagasy government has established a network of PAs. These areas are managed by various institutions, including the state (via the Ministry of Environment and Sustainable Development), Madagascar National Parks (MNP), a private association recognized as a public utility, and both international and national NGOs.

However, PAs in Madagascar face several challenges. These include the difficulty to strengthen collaboration with local communities, the lack of effective management in some areas -commonly referred to as paper parks- and the presence of orphan PAs, which received a legal status but do not have managers. In addition, inadequate funding hampers not only the proper functioning of these areas but also the enforcement of laws [10]. These various challenges manifest differently across PAs, resulting in heterogeneous impacts on their effectiveness throughout the country.

Numerous studies at both global and national levels have shown mixed results of PAs' effectiveness in combating deforestation. The meta-analysis conducted by Börner et al.[11] shows that while protected areas can reduce deforestation, they may also have no effect or even increase deforestation in some cases. Wolf et al.[12] conducted the first comprehensive global assessment of the impact of PAs on deforestation between 2001 and 2018. Their findings indicate that, while PAs contribute to reducing deforestation, they do not entirely eliminate it.

Existing efforts to evaluate the impact of PAs on deforestation in Madagascar have utilized varied techniques and produced divergent or inconclusive results. Discrepancies arise from inadequate data, differing impact estimation methods, and variations in research scopes and statistical specifications. Gorenflo et al.[13] proposed the first structured systematic approach based on remote sensing from 1990 to 2000 to show that PAs reduced the risk of an area being deforested by only 5%.

Desbureaux et al.[14] used the same forest cover and loss data than our study, but the data availability period was then restricted to 2001–2012. They focused on the humid forests of the eastern part of the island and adopted an identification strategy

based on propensity score matching, controlling for socioeconomic factors and policy effectiveness at the municipal level. Their results indicated a 20% reduction in deforestation attributable to PAs. However, in a complementary unpublished work focusing on the same region and timeframe, Desbureaux et al.[15] showed how PAs displaced deforestation into the surrounding landscape, partially offsetting the observed reductions in deforestation inside the PAs.

Another study assessed the impact of PAs on deforestation across two decades, 1990–2000 and 2000–2010, and examined the three main forest types in Madagascar (humid, dry, and spiny). This research controlled for distance to roads, rivers, major cities, altitude, slope, and annual rainfall, using a multivariate ordination approach [16]. This counter-factual method aims to maximize the similarity of treatment and control areas and their estimations find that PAs reduce deforestation. They compared deforestation outcomes between matched controls and treatment pixels, on the one hand, for the period 1990–2000 and, on the other hand, for the period 2000–2010, and find that the estimated impact of PAs on deforestation differs across forest types and periods. In humid forests, the counterfactual difference in deforesta-tion rates is smaller in 2000–2010 than in 1990–2000, whereas it increased in dry and spiny forests. This study relied on forest cover maps for 1990, 2000, and 2010 produced by Conservation International and disseminated by the National Office for the Environment [17]. More recent studies rather use the Hansen et al.[18] data, which results from a forest remote sensing and classification algorithm performed along ecologically defined strata, using normalized multi-date data, and is considered as providing a more reliable assessment of changes in forest cover.

In sum, existing evaluations of the national-scale impact of PAs on deforestation in Madagascar provide useful information but are limited using an incomplete set of control variables, or a too short timeframe to capture significant variation in PA expansion. The objective of this study is to evaluate the influence of PAs on forest cover change in Madagascar over the period 2000–2023 and overcome these limitations by utilizing a deforestation dataset spanning 24 years and employing rigorous impact evaluation methods for a national-scale analysis. The combination of Coars-ened Exact Matching (CEM) and Difference-in-Differences (DiD) creates a robust counterfactual framework that reduces methodological limitations of earlier studies. Matching improves comparability between treatment and con-trol groups by ensuring similar distributions of covariates. CEM is a specific matching method based on the principle of monotonic imbalance bounding [19,20] which means that the researcher can fix ex-ante the maximum imbalance between the treated and the control group for each variable, and that the coarsening choice of one variable does not influence the imbalance bound for the other variables [20]. Additionally, this method does not require any ex-post balance checks [20] nor assumptions regarding how the data was originally collected [20,21]. Empirical evidence suggests that CEM consistently produces superior balance between treated and control groups compared to alter-native matching methods [21,22]. However, matching alone does not provide impact estimates [23]. Therefore, we will use a DiD approach to estimate treatment effects. DiD is commonly employed to identify the causal effects of policy interventions. The canonical DiD approach identifies the average treatment effect under the assumptions of no anticipation and parallel trends [24]. Unlike previous research, this study will also account for spillover and leakage effects and conduct a PA-specific analysis, allowing for a more nuanced understanding of how individual PAs impact forest cover loss. We will employ forest definitions adapted to the very different ecosystem in Madagascar, following the Tree Canopy Density threshold (TCD) criteria developed by Rafanoharana et al.[25]. By addressing both meth-odological gaps and the heterogeneity of PAs, this research seeks to provide a clearer and more reliable estimate of the effectiveness of PA in Madagascar, thereby contributing significantly to conservation science and policy discussions.

## Materials and methods

### Pre-analysis plan objectives

The pre-analysis plan (PAP) is a document developed prior to data analysis, which details the observations and variables to be included in the analysis, as well as the statistical models to be applied, with their specifications. It serves several

functions, as outlined by Olken [26]. First, it helps protect against the conscious or unconscious temptation to adjust statistical specifications based on desired results. Additionally, the PAP improves the conceptual rigor of the study by establishing a clear methodological roadmap. Moreover, it promotes transparency and reproducibility.

For this study, the PAP will guide the analysis of the impact of PAs on deforestation, ensuring that the methodology and data used are appropriate and consistent with the research objectives. This approach helps mitigate risks like p-hacking (bias through pre-specified models) and cherry-picking of results, thus boosting the transparency and credibility of the research. It also sets clear expectations among co-authors by aligning objectives before the analysis begins. The next steps involve submitting the PAP as a registered report protocol to a journal welcoming this type of contribution, then collecting the data from the sources mentioned below, process them, report and interpret the results.

We reserve the right to modify the analysis protocol if, after conducting a descriptive analysis of the data, we find inconsistencies with the estimation methods. The changes made will be timestamped, and each version will be uploaded to ensure transparency in our approach.

## Design

This study is based on Rubin's causal model of potential outcomes [27] to assess the impact of PAs on deforestation. According to the counterfactual framework, for each unit (in this case, a geographical area), there are two potential outcomes: one in which the unit receives the treatment (protection) and one in which it does not. Specifically, the two possible outcomes are the loss of forest cover in the presence of a PA, $Y_1$, and the loss of forest cover if the area is unprotected, $Y_0$. The causal effect of protection on deforestation is the difference between these two potential outcomes. However, in practice, only one of these outcomes can be observed, while the other remains hypothetical. The challenge of causal inference lies in estimating this unobserved counterfactual. In this study, matching techniques are employed to find comparable untreated areas (control units) that are similar to the treated areas on key characteristics which influence the probability of receiving the treatment (i.e., protection) and outcomes (i.e., deforestation). DiD will be used to compare the outcomes between treated and control units.

## Treatment assignment

In its 2008 guidelines, the International Union for Conservation of Nature (IUCN) defines a PA as: "a clearly defined geographical space, recognized, dedicated, and managed, through legal or other effective means, to achieve the long-term conservation of nature with associated ecosystem services and cultural values."[28].

In this study, the treatment units are all terrestrial PAs formally gazetted between 2002 and 2022 with a defined IUCN category. The existence of these PAs must be justified by a provisional creation order or definitive creation decree. The control units will be unprotected areas which are as similar as possible in terms of observable characteristics (accessibility, elevation, slope, initial forest cover, and biome) to the PAs.

## Observation units

The units of observation in this study will be spatial cells with an area of 1 km$^2$, which we have artificially created to analyze deforestation patterns. The selection of cell size is crucial due to the Modifiable Areal Unit Problem (MAUP), as highlighted by Avelino et al.[29]. MAUP can introduce bias into the estimates of program effects if the spatial scale of analysis does not align with the scale at which covariates are available and land-use decisions are made. Our data encompasses resolutions ranging from about 1 ha to 1 km$^2$. Understanding the processes driving deforestation helps determine the most appropriate spatial scale for its measurement and analysis.

The primary driver of deforestation in Madagascar is agriculture [16,30]. While cash cropping, construction of new roads, and large-scale commercial plantations also contribute significantly to forest loss [31], shifting or swidden agriculture remains the predominant livelihood activity practiced by Malagasy rural households to meet their subsistence needs in forested areas. On

average, farms in Madagascar cover an area of 0.87 ha in size [32]. However, using a spatial scale corresponding to this farm size is not suitable for this study, as some covariates used are only available at a lower resolution. For instance, population density and accessibility data are available at a 1 km² resolution. Additionally, forest cover's data and topographic variables such as slope and elevation are obtained at a finer resolution of 0.09 ha. Decisions to change land use are not individual decisions either but may be influenced by the decision of the neighboring landowner due to spatial correlation. They depend on the socio-economic context as well as the spatial dynamics of the region [33]. To address the potential biases associated with MAUP, we will conduct robustness checks, comparing our results with 1 km² cells with results at a different scale of 1 ha, as recommended by Avelino et al.[29].

## Sampling plan

**Source of data used for the study.** This study is based on spatial data describing geophysical characteristics of observation units, attributes of PAs, forest cover. This section also describes tools and packages to be used for the analysis.

**Data on the geographical characteristics of analysis units:** We will rely on the mapme.biodiversity R package to fetch geographical data and compute spatial indicators [34]. It streamlines the acquisition and processing of freely accessible geospatial data related to biodiversity and was specifically designed for monitoring and evaluating conservation programs. One of its advantages is that it provides a list of the most recognized sources in the literature, retrieves them automatically from their reference source, and applies state-of-the-art methods to calculate spatio-temporal indicators derived from these sources. It facilitates the processing of large datasets by utilizing efficient computational routines and parallel computing. The data collected through this channel include the following:

- The ecoregion data is derived from the work of Olson et al.[35]. The publication Terrestrial Ecoregions of the World, made available by the World Wide Fund (WWF), provides a global map of terrestrial ecoregions.

- The data on topographic variables comes from NASA Digital Elevation Model (NASADEM). The image resolution is one arcsecond, equivalent to 30 m at the equator.

- Climate and weather data are available at 1 km² of resolution from WorldClim. It is a dataset of global climate data provided by Fick and Hijmans [36] from 1960 to 2024 at a monthly basis.

- The WorldPop organization provides an estimate of population numbers from the year 2000 at a spatial resolution of 1 km². The mapme.biodiversity package calculates the total population for each observation unit grid. These datasets are developed by WorldPop and CIESIN [37].

- Datasets on accessibility are from the Joint Research Center of the European Commission. They were developed by Uchida and Nelson [38] and offer information on accessibility to cities with populations of at least 50,000 people, starting from the year 2000. The spatial resolution of this dataset is 1 km².

- The cyclone exposure is obtained from the International Best Track Archive for Climate Stewardship (IBTrACS) which merge recent and historical tropical cyclone data from 1984 until now.

**Data on protected areas:** Most studies on PAs rely on the World Database on Protected Areas (WDPA). It is considered as one of the most comprehensive global registers for marine and terrestrial PAs [37,39]. For our study, we are going to use the recent available data on WDPA for Madagascar. The government's initiative to expand the network of protected areas may result in two protected areas overlapping or a new protected area being located within the buffer zone of an older protected area. In such cases, the treatment year is assigned as the year the older park was created to ensure that there is only one treatment.

In Table 1, the year 2002 is considered a pivot year, serving as the reference point for identifying waves of PA creation. There are a total of 138 PAs with status in Madagascar according to the data from WDPA in 2025. Of these, 109 were established after 2002 and 29 before 2002. Among the 109 PAs, 91 are terrestrial, covering an area of 55,989 km², 11 are marine, covering an area of 14,106 km² and 7 mixed PA of 3,862 km². Before 2002, 29 PAs were established covering an area of 16,706 km².

**Table 1. Number of PAs with status by periods of creation.**

| Period of creation | Number of protected areas | | | | Area (km²) | | | |
|---|---|---|---|---|---|---|---|---|
| | Terrestrial | Marine | Mixed | Total | Terrestrial | Marine | Mixed | Total |
| After 2002 | 91 | 11 | 7 | 109 | 55989 | 14106 | 3862 | 73957 |
| Before 2002 | 28 | 0 | 1 | 29 | 15145 | 0 | 1561 | 16706 |
| Total | 119 | 11 | 8 | 138 | 71134 | 14106 | 5423 | 90663 |

Source: Calculations from the authors based on data from WDPA, October 2025.

**Data on forest cover:** The forest cover data used in this analysis is from Global Forest Watch (GFW) platform which provides datasets produced by Hansen et al. [18]. These datasets, derived from Landsat 5, 7, and 8 satellite imagery, have a spatial resolution of 30 m. The tree cover data is available from the year 2000 onwards, while data on tree cover loss begins in 2001.

## Data processing instruments

The data thus obtained will be analyzed and processed on R.

One of the packages we will use is the MatchIt. Following the recommendations of Ho et al.[40], it facilitates the estimation of treatment effects by pre-processing data through semi-parametric or non-parametric matching methods.

The lines of code used for the analysis will be stored in a GitHub repository and on Software Heritage https://www.softwareheritage.org/. GitHub is a cloud-based platform that allows users to store code, manage changes over time, and collaborate with others by sharing and updating code. This will enable modifications to be tracked and simplify collaboration with co-authors. The GitHub repository also facilitates verification by reviewers and reproduction of calculations by interested researchers. As for the data, it will be accessible through the DataSuds portal (https://dataverse.ird.fr/).

## Empirical analysis

### Variables used in the analysis.

**Outcome variable:** The outcome variable is a variable that measures the impact of a treatment. In this study, it will be the percentage loss of forest cover, obtained from data in Hansen et al.[18] and is defined as "stand-replacement disturbance or the complete removal of the tree canopy at the Landsat pixel scale"[18].

We will use Hansen et al.'s data to determine initial forest cover in the year 2000. The annual percentage of deforestation will be computed for each grid cell, as the share of its forested area at year n that is lost by year n + 1.

However, the data from Hansen et al.[18] rely on a standard Tree Canopy Density (TCD) threshold of 30%, which may bias deforestation estimates. According to Rafanoharana et al.[25], this uniform threshold tends to underestimate the extent of dry forest and overestimate humid forest. Specifically, under the 30% threshold, some dry forest areas could be incorrectly classified as deforested (because their TCD is naturally below 30%), whereas humid forest areas might incorrectly appear forested (due to a TCD above 30%), despite actual deforestation having occurred. To address this bias, we will compute specific TCD threshold for each simplified forest type, following the approach of Rafanoharana et al.[25]. The simplified forest type classification is based on the TEOW data mentioned above. Based on the work of Rafanoharana et al.[25], we will then apply the specific threshold for each forest type to the Hansen et al. [18] tree canopy cover map for the year 2000 to create a map of baseline forest cover that is tailored to each forest type.

**Matching:** Matching is a non-parametric data pre-processing method used to control for confounding factors, which are the characteristics influencing both treatment exposure and outcome, to construct a counterfactual [20,41]. The primary goal of matching is to balance covariate distribution between treated and control units to ensure comparability [40,42]. In this study, the treated units are 1 km² grid cells within PAs and the control units are grid cells in unprotected

areas outside the boundaries of any PA. PAs are known to be non-randomly distributed since they have been implemented in remote places which discouraged deforestation [43,44]. Their location can bias the assessment of their impact, so it is mandatory to control for factors related to it such as slope, elevation, accessibility [44]. Among the various matching methods available, we employ CEM to construct the counterfactual for the PAs. CEM is a computationally efficient and conceptually straightforward procedure that improves the balance between treated and control groups by explicitly defining in advance the acceptable level of imbalance [45]. This approach reduces the need for repeated balance checks and minimizes the model dependence in the estimation process.

**Matching variables:** According to Ho et al.[40], matching covariates are the variables that affect both the treatment assignment and the outcome. In our context, these covariates primarily reflect the geographic location of PAs and also influence deforestation, allowing us to control for location-specific biases. Moreover, we have included few pretreatment covariates based on our background knowledge:

- Accessibility to cities in 2000: This variable is defined as the duration, in minutes, required to reach a city with a population of 50,000 inhabitants or more, using data available from 2000 based on Uchida and Nelson [38]. This variable is pertinent to this study because it helps correct for bias due to the remote placement of PAs and the fact that remote areas are less likely to be deforested [43,46].

- Population density: This variable corresponds to the estimated number of inhabitants per $km^2$ based on the WorldPop data described above. Literature identifies threats to biodiversity resulting from anthropogenic pressures [46,47]. PAs have also been located in areas with lower population density [43]. Therefore, it is essential to include this variable in the matching process to pair areas with similar population sizes taking the year 2000 as reference.

- Simplified forest types: This variable indicates the type of forest, following the grouping of ecoregions used by Rafanoharana et al.[25]. It classifies Madagascar into four simplified forest types: Humid, Sub-Humid, Dry, and Spiny. These correspond to groupings available in the ecoregion data mentioned above. Note, however, that mangroves are present in the ecoregion data but missing in this simplified classification, and we will separate mangrove areas for the analysis.

- Slope and elevation: PAs were not randomly placed [4]. They are generally established in areas with the lowest opportunity costs, typically at higher elevations and on steeper slopes, which are less suitable for resource extraction or agricultural activities [43,44]. Their location can bias the estimation of their impacts, as these areas could have had a low percentage of forest cover loss even without protection [4]. Elevation and slope are topographical indicators that help locate PAs. They can be obtained from NASADEM data.

- Forest cover in 2000: This variable corresponds to the percentage of canopy cover of the 1 $km^2$ grid cell in 2000. It is crucial for the study as it will serve as a reference year for forest cover, enabling the measurement of the initial characteristics of the units being compared. The year 2000 is chosen because it is the first year of availability of Hansen et al.[18] data, from which it is extracted.

**Control variables:** Control variables are covariates unrelated to treatment assignment yet capable of influencing the outcome. Including them in the regression reduces bias in the estimates and while not strictly required, enhances the precision of the estimated causal effect of PAs on deforestation [48]. The variables listed below were measured at an annual resolution and will serve as control variables in the model:

- Drought indicator: Desbureaux and Damania [49] state that Malagasy agriculture is 80% rain-fed. Drought can be defined as a long period without precipitation [50]. Lack of precipitation can be a driver of deforestation [51] as the resulting drop in agricultural yield encourages farmers to increase the area of cultivated land [49]. The Standardized Precipitation Evapotranspiration Index (SPEI) will be used to control for the drought phenomena as it is multiscalar, which helps to analyze and monitor drought [50]. It will be calculated from the total monthly precipitation data, along

with the minimum and maximum monthly temperature data available on WorldClim, following the improved Hargreaves method defined by Droogers and Allen [52]. It will be computed annually, with a three-year lag to capture delayed ecological responses.

- Cyclone exposure: According to Andrianambinina et al.[53], tropical cyclones can be a factor in deforestation in Madagascar, depending on the degree of exposure of the area. This indicator can be obtained using data from IBTrACS. It will be calculated each year from 2000, with a one-year lag to capture the effects of cyclones from the previous year on forest cover in year (t).

## Analysis plan

**Matching.**

**Coarsened exact matching:** CEM is a matching method that consists of distributing covariates into different bins and matching treatment units and control units that are in the same bin. Bins are intervals obtained from coarsening the value of covariates by dividing them into equal size groups using quantiles as thresholds [12]. In CEM, treatment and control units are assigned to bins based on the coarsened values of all their covariates. Units that fall into the same bin are matched because they share identical coarsened characteristics. Bins that contain only treatment units or only control units are excluded from the analysis because they lack units from the opposite group. The use of CEM requires addressing a trade-off between minimizing imbalance between treatment and control units and retaining a sufficient number of treatment units. Prioritizing balance may result in the exclusion of more units, while prioritizing the retention of units may increase imbalance.

**Decision algorithm:** We will define the number of quantiles using a procedure designed to reach a target Standardized Means Difference (SMD) between treatment and matched controls of 0.25 for all continuous variables, while aiming at excluding less than 10% of treated units. If the target SMD cannot be achieved without exceeding 10% attrition, the tolerance will be increased up to 20%, and both the estimate maximizing comparability and the estimate maximizing representativeness will be reported. The detailed procedure is as follows:

- The starting point will be the coarsening of continuous variables into two bins. At this stage, we expect limited similarity between the treatment and control groups, with a SMD largely superior to 0.25 for most continuous variables.

- From this starting point, we will iteratively increase the bin number by one for all variables with an SMD > 0.25, until reaching an SMD ≤ 0.25 for all continuous variables.

- Each time we increase the number of bins, we will check the number of treatment units outside the common support area, as these must be removed from the analysis. If more than 20% of the treatment units are removed, we return to the previous iteration which has removed less than 20% of the treatment units and stop the iterations.

- For any protected area where the above procedure leads to the exclusion of more than 10% of treated units, we will also report an alternative estimate based on a coarser binning (with higher SMD, > 0.25) that retains at least 90% of treated units. This dual reporting will provide a transparent presentation of the trade-off between comparability (covariate balance) and representativeness (sample coverage).

If there are not enough control units to match with the treatment units using CEM, we will employ kernel matching as an alternative. This method will assign weights to all control units based on their distance to treated units, using a Gaussian kernel. The bandwidth parameter, which determines the weight distribution, will be optimized to achieve balance between treatment and control groups. To ensure the robustness of our results, we will conduct sensitivity tests using nearest neighbor matching with alternative specifications.

### Difference-in-differences estimation.

**Difference-in-differences with Two Way Fixed Effects:** The DiD method is a research design used to evaluate the causal effects of a treatment when it has not been assigned randomly [23]. It is based on comparing the average change in outcomes in the treated group to the average change in outcomes in the control group [24,54]. The DiD relies on the parallel trends' assumption, which supposes that treatment and control units show the same trend in outcomes before the intervention, meaning it is reasonable to assume that they would have continued to follow the same trend in the absence of the intervention [55,56]. The canonical form of the DiD involves a single treatment period with one treatment group and one control group. Including fixed effects helps control for unobserved variables that are constant over time, thus improving the validity of the estimation [54].

$$Y_{it} = \tau D_{it} + \mu_i + \lambda_t + \beta X_{it} + \varepsilon_{it} \tag{1}$$

Where:

$Y_{it}$: Percentage loss of forest cover in unit i at a time t

$D_{it}$: Interaction term between treatment and period, equal to 1 if unit belongs to the post treatment group and 0 if it belongs to the control or pre-treatment group

$\tau$: Coefficient of interest

$\mu_i$; Unit fixed effect

$\lambda_t$: Time fixed effect

$X_{it}$: Control variables

$\varepsilon_{it}$: Error term

However, this model may not be suitable for PAs, as they can produce potential spillover and leakage effects. It is therefore necessary to adopt another model that can consider these spatial dynamics.

**Difference-in-differences with spillover and leakage effects:** The protection of an area can have impacts that extend beyond its boundaries and affect nearby, unprotected areas. When benefits of a treatment cross its borders, this is referred to as spillover effects [57]. In contrast, leakage effect occurs when the protection displaces deforestation to adjacent areas [58]. In this study, spillover effects refer to reductions in deforestation in adjacent areas, whereas leakage effects correspond to increases in deforestation near PAs.

To assess these effects, we follow the method developed by Butts [57], which involves determining the maximum distance beyond which the spillover and leakage effect becomes negligible, and assessing their intensity as a function of fractions distance from PA boundaries. To do this, this distance is divided into several concentric rings.

For the purposes of this analysis, we consider the maximum distance of potential occurrence of the spillover and leakage effects to be 20 km. Most studies of PAs use a 10 km buffer zone [4,59], but we widened this to 20 km to allow for a broader analytical scope. This distance will be divided into three rings following Butts' multiple rings approach:0–5 km, 5–10 km, and 10–20 km, with each ring including its maximum limit. The units located inside each ring will be compared to control units located beyond 20 km from the PA. This methodology enables us to estimate two distinct estimands with a single specification: the total effect of protection on treated units within the PA and the spillover or leakage effects in surrounding areas.

Following Butts [57], the total effect of protection on the treated can be first expressed as:

$$T_{total} = E\left[Y_{i1} - Y_{i0}/D_i = 1\right] - E\left[Y_{i1} - Y_{i0}/D_i = 0, S_i = 0\right] \tag{2}$$

Where:

$\tau_{total}$: Total treatment effect

$Y_{i1}$: Potential outcome after treatment

$Y_{i0}$: Potential outcome before treatment

$D_i$: Treatment indicator which is equal to 1 if unit i is treated and 0 if the unit i is untreated

$S_i$: Exposure indicator equal to 0 if the unit is located beyond 20 km of a PA

This formulation identifies the total effect on treated units by comparing them with unexposed control units located more than 20 km away, assumed to satisfy the parallel-trends assumption.

To estimate both the treatment and the spillover effects jointly, Butts [57] extends this specification as follows:

$$Y_{it} = \alpha + \tau D_{it} + \gamma (1 - D_{it}) S_i (\overline{d}) + \beta X_{it} + \mu_i + \lambda_t + \varepsilon_{it} \tag{3}$$

Where:

$Y_{it}$: Percentage loss of forest cover in unit $i$ at time $t$

$\alpha$: Intercept

$\tau$: Coefficient of interest representing the total treatment effect on treated units (inside PAs)

$\gamma$: Coefficient representing the average spillover or leakage effect on non-treated units located within 20 km of the PA

$D_{it}$: Treatment indicator equal to 1 if unit i belongs to the post-treatment group and 0 if it belongs to the control or pre-treatment group.

$(1 - D_{it})$: Indicator for untreated units in the buffer zone

$S_i (\overline{d})$: Spillover exposure indicator which is equal to 1 if unit $i$ is within the buffer zone of 20 km and 0 otherwise

$\beta X_{it}$: Vector of control variables

$\mu_i$: Unit fixed effect

$\lambda_t$; Time fixed effect

$\varepsilon_{it}$: Error term

By decomposing this single ring into several concentric rings, the equation becomes:

$$Y_{it} = \alpha + \tau D_{it} + \sum_{j=1}^{3} \gamma_j (1 - D_{it}) Ring_{ij} + \beta X_{it} + \mu_i + \lambda_t + \varepsilon_{it} \tag{4}$$

The terms of the equation are defined as follows:

$Y_{it}$: Percentage loss of forest cover in unit $i$ at time $t$

$\alpha$: Intercept

$\tau$: Total effect on treated units

$\gamma_j$: Spillover or leakage effect for ring j, measured relative to unexposed control units located beyond 20 km

$D_{it}$: Treatment indicator equal to 1 if unit $i$ belongs to the post-treatment group and 0 if it belongs to the control or pre-treatment group.

$(1 - D_{it})$: Indicator for untreated units

$Ring_{ij}$: Indicator that unit $i$ belongs to ring $j$

$\beta X_{it}$: Vector of control variables

$\mu_i$: Unit fixed effect

$\lambda_t$: Time fixed effect

$\varepsilon_{it}$: Error term

In this unified specification, both treated units ($D_{it} = 1$) and buffer units ($D_{it} = 0$, $Ring_{ij} = 1$) are compared to a common reference group: control units located more than 20 km from any PA boundary. For treated units, the spillover terms drop out and $\tau$ identifies the total effect of protection; for buffer units, the treatment term equals zero and $\gamma_j$ identifies the spillover or leakage effect in each distance ring. These parameters are jointly estimated but conceptually different: $\tau$ is the total

effect on treated units, while $y_j$ corresponds to the spatial externalities on nearby controls. Both will be reported separately in the analysis.

### Inclusion and exclusion criteria

In this study, only terrestrial PAs will be considered, including the terrestrial portion of coastal areas, but excluding marine PAs and the marine portion of coastal areas. Furthermore, areas covered by a management status other than a status of official PA (that is, areas without a provisional order or definitive PA creation decree) will not be considered as treated and will be considered as potential controls instead.

PAs created before 2002 will not be studied or used as control areas.

Units located within the 20 km buffer zone will be excluded from the control group when matching with units inside PAs. Instead, they will be used for the spillover and leakage analysis by matching them with more distant control units.

### Iterative analysis by protected area

We will proceed iteratively, computing the impact estimation for each PA individually. All empirical analyses will be applied recursively to each PA, with parameters specific to each PA being recalculated. To classify each area as «forest » and « non-forest », we will use the mean TCD of PAs belonging to the same simplified forest type, as defined by Rafanoharana et al [25]. We will then apply this threshold across the entire biome of each PA to determine forest and deforestation status.

### Robustness test

The specification of the model used in the above matching is based on a review of the literature on the subject in question. To check the robustness of the model, we need to:

- Retest the same model on analysis units of 1 ha to measure the differences caused by the choice of scale.

- Mask the Hansen et al.[18] data with the Harper et al.[17] forest cover data to identify differences in results caused by discrepancies between the two maps.

- Assess the robustness of estimates for protected areas with limited common support. For cases where we could not achieve an SMD ≤ 0.25 with less than 10% of excluded treated units, we will compare the main estimate (maximizing covariate balance) with an alternative estimate based on a coarser binning (retaining at least 90% of treated units). This comparison will assess the sensitivity of results to the balance–representativeness trade-off in the matching procedure.

- Perform the sensitivity test for unobserved bias according to Rosenbaum [60], which can be carried out using the R package rbounds. The covariates we choose to include in the model can affect our findings [61]. In observational studies, we match treated and control groups on observed covariates, but they may still differ on unobserved factors. Those unseen differences can influence the outcome, making it impossible to attribute any effect solely to the treatment [62]. When two units are identical with respect to a variable x, they should have the same probability of receiving the treatment. The presence of unobserved bias can lead to a difference in this probability, which is measured by the parameter Γ. The sensitivity test consists in gradually increasing the value of Γ to determine the point at which the results of the estimates are modified. The model is sensitive if for a value of Γ close to 1, the estimates change. The model is robust if a high value of Γ is required to affect the results.

## Discussion

### Heterogeneous treatment effects

This study focuses on the impact on deforestation of PAs created between 2002 and 2022. The result of this study will be a collection of impact estimates with one estimate per PA over the study period. Subsequently, we will analyze the

distribution of these impact estimates per PA, to identify heterogeneity and patterns that might explain this heterogeneity. We will, for instance, analyze the impact estimate variation per IUCN category, size or type of management organization. This will enable us to identify which IUCN category is most effective in maintaining forest cover. In addition, analysis of the location of PAs, by biome, type of crops cultivated in the area (e.g., vanilla, cocoa, maize), and administrative region will indicate if differences appear regarding these specifications. We will further examine the fluctuation of crop prices over time to assess their potential influence on deforestation dynamics in future research.

### Future research

We expect to find strong heterogeneity and patterns related to time, space and PA types. These patterns will be the starting point of future research, which will not be addressed in this paper, but the discussion section of this paper will frame and orient these future inquiries. We anticipate that such future research will adopt a mixed methodology combining quantitative and qualitative methods to understand the determinants of observed results and their variability. In particular, the aim will be to understand how the effectiveness of PAs is determined by local socio-economic and political contexts, PA financing and management practices, as well as local governance of the protected and surrounding areas.

### Supporting information

**S1 Table.** It summarizes the different variables used in the analysis. The selected articles applied matching method to assess the impact of conservation policies and addressed similar research questions. The table includes 11 studies, including 6 carried out in Madagascar. While not an exhaustive review of impact evaluations of conservation programs in Madagascar, it provides an overview of the variables taken into account to find the protected areas counterfactual given the main drivers of deforestation.
(DOCX)

### Acknowledgments

This research is co-funded by the Development Impact Lab of the German KfW Development Bank, the French Development Agency (AFD) through the PAIRES program and the French National Research Institute for Sustainable Development (IRD). We would also like to express our gratitude to all those working in the field of conservation in Madagascar for their advice and ideas, without which this research would not have been possible.

### Author contributions

**Conceptualization:** Diamondra Ramiandrisoa, Florent Bédécarrats, Thierry Razanakoto.

**Data curation:** Diamondra Ramiandrisoa, Florent Bédécarrats.

**Formal analysis:** Diamondra Ramiandrisoa, Florent Bédécarrats, Melvin H.L. Wong.

**Funding acquisition:** Florent Bédécarrats.

**Investigation:** Florent Bédécarrats.

**Methodology:** Diamondra Ramiandrisoa, Florent Bédécarrats, Melvin H.L. Wong.

**Project administration:** Florent Bédécarrats.

**Supervision:** Florent Bédécarrats, Thierry Razanakoto.

**Writing – original draft:** Diamondra Ramiandrisoa, Florent Bédécarrats.

**Writing – Review & Editing:** Diamondra Ramiandrisoa, Florent Bédécarrats, Melvin H.L. Wong.

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
