## [Decision Letter · Decision Letter 0]

9 Jun 2025

Dear Dr. Ramiandrisoa,

We look forward to receiving your revised manuscript.

Kind regards,

Daniel de Paiva Silva, Ph.D.

Academic Editor

PLOS ONE

Journal Requirements:

2. In your cover letter, please confirm that the research you have described in your manuscript, including participant recruitment, data collection, modification, or processing, has not started and will not start until after your paper has been accepted to the journal (assuming data need to be collected or participants recruited specifically for your study). In order to proceed with your submission, you must provide confirmation.

“The project is funded by KfW, a financial institution that provides funding for protected areas in Madagascar and other countries. One of the authors of this paper is employed by KfW as a project manager in the evaluation department. The evaluation department was designed to be independent: it is directly subordinate to the Executive Board of the entire KfW Group, works independently of the operative departments of KfW Development Bank and is managed by a person from academia, designed by the KfW Executive Board.”

We note that you received funding from a commercial source: [Name of Company]

Additional Editor Comments (if provided):

Dear Dr. Ramiandrisoa,

After this first review round, both reviewers believe your manuscript has a significant methodological potential for future studies in other regions. Still, one of the reviewers raised important issues that need to be accounted and addressed before the text is accepted for publication in PLoS One. Specifically, this reviewer raised issues related to improvements regarding the statistical analyses you employed and also a better evaluation involving the selected variables dataset related to the deforestation estimates. In addition, several minor suggestions have been made.

I believe that if you are able to convince the reviewers in the next review round, your manuscript will be ready for publication after major reviews.

Sincerely,

Daniel Silva

Reviewers' comments:

Reviewer's Responses to Questions

**Comments to the Author**

1. Does the manuscript provide a valid rationale for the proposed study, with clearly identified and justified research questions?

Reviewer #1: Yes

Reviewer #2: Yes

2. Is the protocol technically sound and planned in a manner that will lead to a meaningful outcome and allow testing the stated hypotheses?

Reviewer #1: Yes

Reviewer #2: Partly

3. Is the methodology feasible and described in sufficient detail to allow the work to be replicable?

Reviewer #1: Yes

Reviewer #2: No

4. Have the authors described where all data underlying the findings will be made available when the study is complete?

Reviewer #1: Yes

Reviewer #2: No

5. Is the manuscript presented in an intelligible fashion and written in standard English?

*PLOS ONE*

Reviewer #1: Yes

Reviewer #2: Yes

You may also provide optional suggestions and comments to authors that they might find helpful in planning their study.

Reviewer #1: This report meets the criteria for the Registered Report Protocol publication format by describing the state of knowledge, research questions and methodology for the future development of a research proposal. The results of this research proposal will have an impact on the conservation of protected areas in tropical regions by providing insights into predicting the impacts of forest loss on protected areas. I have a few observations:

L100-102. In this section, it is necessary to provide a brief background on the benefits of using Coarsened Exact Matching (CEM) and Difference-in-Differences (DiD).

L110-128 This section should be part of the introduction or materials and methods.

Reviewer #2: This is a very impressive, ambitious and much-needed study you are planning. I especially liked the idea of estimating the individual effect of each PA and mapping this to explore spatial patterns of effectiveness. You definitely can then relate effectiveness to a set of potential explanatory variables but I worry this might be too much for a single study and may be better reserved for a follow-up. I think the proposed methodology is mostly sound but I do have some serious concerns which will need to be rectified for this study to be suitable for publication. In some places the methodology is not clearly explained and is missing critical information needed to assess its feasibility. I have detailed these below. Overall, the paper needs quite a substantial re-write to ensure the methodology is clearly explained, justified and consequently replicable.

Main points:

1) Here, or in the Introduction to the main Registered Report, I recommend you discuss some of the challenges facing conservation and Protected Areas in Madagascar: paper parks, lack of funding, lack of income from tourism in the majority of PAs, corruption, insecurity (daholos etc.), lack of enforcement capacity. This is really important framing for your study, emphasizing why it is important to evaluate the effectiveness of PAs individually, so you can try and understand what factors are associated with PA successes or failure.

2) In the Introduction you suggest that the Hansen Global Forest Change data is better to use than the Harper et al forest cover maps for measuring forest cover change. You talk about the limitations of the Harper et al., data (which come from the Eklund et al., paper and not the original source) but do not mention at all the known limitations of the GFC data and other global products. This is unbalanced and gives the impression that the Harper et al., data is ‘wrong’ and the Hansen GFC data is ‘correct’. In my experience, this is not the case. I’ve used the Harper et al maps, the Vieilledent et al., 2018 (https://www.sciencedirect.com/science/article/pii/S0006320718301125), the Global Forest Change data (Hansen et al., 2013) and the more recent Tropical Moist Forests (Vancutsem et al., 2022) data for studying deforestation in Madagascar. Both the Hansen data and the TMF data seem to substantially over-estimate deforestation in Eastern Madagascar, showing massive deforestation between the eastern edge of the CAZ and the east coast. However, the forest cover maps of Harper et al., suggest that these areas were not forested at baseline in 2000.

Personally, I prefer the approach taken by Vielledent et al., 2018. They combined the Harper et al., maps with the GFC data, masking the GFC forest loss data to the map of forest cover in 2000 from Harper et al.,. In the Eastern rainforests, this masking process seems to remove many false positives, where deforestation is detected in places which have not been forested for decades before 2000. I have used the same method (masking global deforestation datasets to the national map for the year 2000) in two previous impact evaluation studies of deforestation in Madagascar. I think this is justifiable for the following reasons. The GFC data is a global dataset so the classification algorithm has to be broad enough to capture deforestation in all the different forest ecosystems on earth. As such, it sacrifices local accuracy for global generalisability. The GFC is also known to be less accurate in capturing dynamics of dry forests. In contrast, the Harper et al., maps were designed and tailored to capture the specific ecological characteristics of Madagascar’s forests. However, this dataset doesn’t have a yearly time series and wasn’t explicitly designed to capture changes. In my opinion combining the two datasets gives the most accurate measure of deforestation on land which was actually forested at the baseline in 2000.

I would strongly recommend you mask the Hansen et al data to the 2000 forest cover map from Harper et al., and take a look. If there are any forested areas you are personally familiar with, compare the deforestation shown in these areas with and without masking.

Previous studies have shown that the choice of data to measure outcomes can have a strong effect on the results of impact evaluations and I’ve seen how much of a difference there is between the global deforestation maps and the Harper et al., map for 2000. If you choose to continue using the Hansen et al., 2013 data for your analysis and I am asked to review the Registered Report, I would want to see additional results using the Hansen data masked to the 2000 forest cover map as a robustness check. Without this I would have doubts about any significant results.

3) I don’t think it’s necessary to run a robustness check using cells of 5km2, I think that’s too large and doesn’t correspond to the scale of land use decision making. Personally, I also would use 1ha as the main scale of analysis. For me, alignment with the scale of land use decision making is more important that alignment with the scale of all covariates. At a scale of 1ha, ~10 neighbouring units will fall within the same 1km2 pixel in the population density and accessibility data. Therefore, all these 10 units will have the same value for these variables. I think that’s okay because in reality, the true values of these variables will differ little within 1km2, especially for the distance variable (which will vary by up to 1km). While I personally would use 1ha as the main scale of analysis and 1km as a robustness check, I will leave it to your discretion which one you use as the main scale. I am happy as long as you do run the analysis at 1 ha resolution (as main or additional).

4) Some important information is missing in the methods:

- Are you only going to include units (1km2 pixels) with a minimum baseline % forest cover? If so, please state the minimum baseline forest cover for a unit to be included in the study. If not, you need to do this.

- Are you going to exact match on forest type? If so, please state this somewhere.

5) I have to admit that I’m confused about your leakage analysis. I am not familiar with the Butts paper but you need to make sure your method is sufficiently clear here so that readers do not need to cross-check with another paper. I’ll try to explain what is causing my confusion to help you with this:

Lines 400-402: ‘These new treatment zones will be matched with remote control zones to obtain the magnitude of spillover and leakage effects.’

This suggests that you will run the leakage analysis separately to the main within-PA analysis, setting units within the buffer ring zones as treated (and therefore excluding units within the PAs), matching to similar controls and running the regressions. This is again suggested in Lines 408 when you refer to buffer zone units as the ‘treatment unit’. And in Lines 433 – 435.

If this is correct you need to make it really clear at the start of the ‘Difference-in-differences with spillover and leakage effects’ section that this is an additional, separate analysis to the main within-PA analysis. It is currently not clear and is confusing. In this additional analysis units within concentric buffer zones outside the PA will be assigned as treated, with treatment occurring in the year the adjacent PA was protected. You can also state in the ‘Treatment Assignment’ section, that you will run two analyses, the first using units from within PAs as treated, and the second using units from a 20km buffer zone around each PA as treated.

Additionally, to make it clear that this is a separate analysis, you should also include the regression equation for the main within-PA analysis in the ‘Basic model of Difference-in-differences’. This will make it clear to readers that the two regressions are different (because the basic model doesn’t include the Ring parameter), therefore this is two separate analyses.

Also, regarding the leakage regression equation shown in Line 411, I’m not convinced about the (1-D) x Ring. If my interpretation is correct, we are interested in the interaction between the ring distance and D = 1, to estimate the effect of a treated buffer zone unit being further from the boundary of a PA (but still within the 20km buffer) after the PA is protected. But 1-1 = 0. So by including (1-D) x Ring aren’t you completely removing the Ring term for the treated units you are interested in?

If I am wrong, and the two analyses are not separate, then you need to explain better how treatment is assigned.

6) In the Heterogeneous Treatment Effects section you state that you will obtain an estimate of the effect of protection within each PA for each year post-intervention. Unless you are planning on conducting an event-study type analysis, which I assume you are not because it is not mentioned anywhere in the text, you will not get an estimate for each year post-protection. Your estimates will represent the average annual change in deforestation in the PA across all years post-protection.

With your study design you will be able to assess spatial heterogeneity, the different performance of different PAs, but not temporal heterogeneity. Including the regression equation for your main analysis would make this very clear.

As such, please delete the text referring to assessing changes in performance over time and instead focus on the patterns, and possible causes of spatial variation in effectiveness.

7) You need to state the data and code used in the study will be made publicly available (e.g. on Github or FigShare) after the study is complete.

8) I really like the idea of trying to explain the differential effectiveness of PAs according to a set of hypothesized explanatory factors but I am concerned this might be too much for a single study and may be better reserved for a follow-up.

Detailed comments:

Line 44- 46: The GFW platform uses Hansen et al., (2013) data so directly cite this data source, and delete the reference to Global Forest Watch.

Line 46 – 48: Delete the statement about how Biodiversity can play a crucial role in the developing country as this is too simplistic are currently written. Instead, you can talk about how Madagascar is one of the poorest countries in the world and it’s forests play a crucial role in supporting the livelihoods of its predominantly rural populations by providing ecosystem services.

Line 49: Delete ‘with the creation of Madagascar’s Protected Areas System (SAPM) because otherwise it sounds like Madagascar didn’t have any PAs before 2003.

Line 52-53: Explain what a negative or null effect means, that these PAs did not decrease deforestation and may even have conversely increased it.

Lines 53- 60: I don’t think you can talk about before-after comparisons without describing what they are. I also don’t think you need to go into this much detail here. It is probably sufficient to say that the Wolf study showed that globally PAs established between 2002 and 2017 showed no significant effect overall, probably due to heterogeneity.

Line 65-66: I don’t understand this statement. Are you saying that the imagery used didn’t correspond to the period 1990-2000?

Line 71-72: Start this sentence with ‘However…’. as it describes contrasting/contradictory results. Desbureaux showed that PAs reduced deforestation risk by 20% but this might not have reduced deforestation at the landscape-scale because deforestation was displaced into the surrounding, non-protected landscape.

Line 80: I’m not sure we need this much detail about the estimand they use. But please state exactly what ‘modest results’ means in terms of the amount of deforestation. Does that mean they found small reductions in deforestation within PAs?

Lines 83 – 89: I recommend deleting this. These limitations described in the Harper et al., data come from the Eklund et al paper but seem to be over-emphasized compared to the original source. Also, without comparing multiple data sources to ground truth data you can’t say which one is ‘better’. See comment 2 below.

Line 82- 83: Cite Harper et al., here.

Line 94: Delete ‘inaccurate data’. With these datasets it is very difficult to say which ones are ‘right’ or not, especially at the national scale.

Line 99: Delete ‘state of the art’.

Lines 93-97: Instead of focussing on the limitations of previous studies, who did what they could with the data available at the time, focus on the novel aspects of your study: the longer time-series, the spillover analysis and particularly that you will get both average (across all PAs) and individual treatment effects. That’s cool!

Line 131: Rubins’ causal model of potential outcomes (please add this underlined text).

Lines 153 – 170: Remove the table and reference to Fritz et al., as these findings refer to the whole of the African continent. Instead, try and find specific references about the leading drivers of deforestation in Madagascar. You don’t need to go into much detail. You can just have one or two sentences saying something like: In Madagascar, the largest driver of deforestation is clearance for subsistence agriculture, in some parts clearance for XXX also plays an important role. But make sure you include Madagascar-specific citations.

Lines 199- 201: Unnecessary, you can delete this sentence.

Line 215 – 217 and Lines 220- 221. Same. Delete, this is unnecessary.

Line 230 – 235: As the study was not repeated using different PA boundary data there is no evidence that this affected the main conclusions of the study. Delete

Line 253 - 257: This paragraph is currently a bit too long and repetitive, please shorten and remove unnecessary information (e.g. that Global Forest watch was established by WRI).

Line 259: Just say that the data will be analyzed and processed on R. No need for the extra information.

Lines 273 – 281: It is unclear how you will incorporate ecological differences between forest types into your outcome variable. Please clarify. Will you use the Hansen et al canopy cover dataset for 2000 to set a specific threshold to map forests of each type, and then mask the deforestation data to this 2000 forest cover map? Also, see my main point about combining the Harper et al and the Hansen et al datasets to create an outcome variable.

Line 282: I recommend you add the paragraph explaining matching methods here, before you talk about matching variables. Please clearly explain what are the benefits of matching including specific reference to the need to control for confounding factors associated with both treatment assignment (i.e. where a PA is located) and outcomes. Failure to control for these factors can mean any differences in outcomes between the two samples might be attributable to these factors, and not the PA status at all. Matching can be used to control for confounding factors to ensure both samples are as similar as possible in characteristics which affect treatment assignment and deforestation.

Line 301: Delete ‘for the nearby population’ as this is often not true.

Line 310: I don’t understand the difference between matching variables and control variables. Are you not matching on the control variables? Please explain.

Line 333-334: I disagree with this statement, please delete. In some places forest plots closer to rivers have a higher probability of deforestation as they are more accessible by water. Furthermore, settlements tend to be located closer to rivers also increasing the likelihood of deforestation. The reference supporting this statement is from a case study in India and should not be used.

Line 384: Changes in the evolution after the intervention

Lines 382- 389: Here you need to explain what difference-in-difference is � comparing outcomes before and after an intervention between treatment and control to assess how an intervention changed outcomes in the treated unit over time, compared to the control.

Line 394: Replace ‘fall’ with ‘reduction’.

Line 408: Replace ‘treatment unit’ with ‘buffer zone ring’

Line 411: Should be Y with subscript i and t.

Line 433: Reword to state that units within 20km buffer of a PA are removed from the control sample for the main analysis, and assigned as treated for the leakage/spillover analysis.

Lines 439 – 442: Can you reword this more simply? Are you saying that for canopy cover thresholds for defining forest cover are set based on the forest type within each PA. There are only four forest types so I recommend stating explicitly here what thresholds you will set for each forest type (e.g. X% for the humid forest, Y% for the dry forest etc…).

Line 448: ‘Perform a sensitivity test for unobserved bias according to…’ Please explain more clearly how unobserved bias could impact results via confounding.

Line 458: Delete ‘per year’. See Main Point 4.

**Do you want your identity to be public for this peer review?** For information about this choice, including consent withdrawal, please see our Privacy Policy

Reviewer #1: No

Reviewer #2: **Yes:** Katie Devenish

---

## [Author Response · Author response to Decision Letter 1]

14 Aug 2025

Response to reviewers :

Response to Reviewer #1

This report meets the criteria for the Registered Report Protocol publication format by describing the state of knowledge, research questions and methodology for the future development of a research proposal. The results of this research proposal will have an impact on the conservation of protected areas in tropical regions by providing insights into predicting the impacts of forest loss on protected areas. I have a few observations:

We sincerely thank you for your positive feedback and for your interest in our work.

L100-102. In this section, it is necessary to provide a brief background on the benefits of using Coarsened Exact Matching (CEM) and Difference-in-Differences (DiD)

The benefits of using CEM and DiD have been added from the line 139 to the line 151.

L110-128 This section should be part of the introduction or materials and methods.

The section about the pre-analysis plan objectives has been moved to the “Materials and methods’ section” in line 173.

Response to Reviewer #2

1) Here, or in the Introduction to the main Registered Report, I recommend you discuss some of the challenges facing conservation and Protected Areas in Madagascar: paper parks, lack of funding, lack of income from tourism in the majority of PAs, corruption, insecurity (daholos etc.), lack of enforcement capacity. This is really important framing for your study, emphasizing why it is important to evaluate the effectiveness of PAs individually, so you can try and understand what factors are associated with PA successes or failure.

Thank you for this recommendation. We recognize that the introduction lacked sufficient contextualization. We have revised the pre-registered protocol accordingly and will continue refining it for the main Registered Report. In particular, as you suggested, we have highlighted Madagascar's specific conservation challenges, including governance issues, lack of funding and the widespread prevalence of poverty from line 50.

2) In the Introduction you suggest that the Hansen Global Forest Change data is better to use than the Harper et al forest cover maps for measuring forest cover change. You talk about the limitations of the Harper et al., data (which come from the Eklund et al., paper and not the original source) but do not mention at all the known limitations of the GFC data and other global products. This is unbalanced and gives the impression that the Harper et al., data is ‘wrong’ and the Hansen GFC data is ‘correct’. In my experience, this is not the case. I’ve used the Harper et al maps, the Vieilledent et al., 2018 (https://www.sciencedirect.com/science/article/pii/S0006320718301125), the Global Forest Change data (Hansen et al., 2013) and the more recent Tropical Moist Forests (Vancutsem et al., 2022) data for studying deforestation in Madagascar. Both the Hansen data and the TMF data seem to substantially over-estimate deforestation in Eastern Madagascar, showing massive deforestation between the eastern edge of the CAZ and the east coast. However, the forest cover maps of Harper et al., suggest that these areas were not forested at baseline in 2000.

Personally, I prefer the approach taken by Vielledent et al., 2018. They combined the Harper et al., maps with the GFC data, masking the GFC forest loss data to the map of forest cover in 2000 from Harper et al.,. In the Eastern rainforests, this masking process seems to remove many false positives, where deforestation is detected in places which have not been forested for decades before 2000. I have used the same method (masking global deforestation datasets to the national map for the year 2000) in two previous impact evaluation studies of deforestation in Madagascar. I think this is justifiable for the following reasons. The GFC data is a global dataset so the classification algorithm has to be broad enough to capture deforestation in all the different forest ecosystems on earth. As such, it sacrifices local accuracy for global generalisability. The GFC is also known to be less accurate in capturing dynamics of dry forests. In contrast, the Harper et al., maps were designed and tailored to capture the specific ecological characteristics of Madagascar’s forests. However, this dataset doesn’t have a yearly time series and wasn’t explicitly designed to capture changes. In my opinion combining the two datasets gives the most accurate measure of deforestation on land which was actually forested at the baseline in 2000.

I would strongly recommend you mask the Hansen et al data to the 2000 forest cover map from Harper et al., and take a look. If there are any forested areas you are personally familiar with, compare the deforestation shown in these areas with and without masking.

Previous studies have shown that the choice of data to measure outcomes can have a strong effect on the results of impact evaluations and I’ve seen how much of a difference there is between the global deforestation maps and the Harper et al., map for 2000. If you choose to continue using the Hansen et al., 2013 data for your analysis and I am asked to review the Registered Report, I would want to see additional results using the Hansen data masked to the 2000 forest cover map as a robustness check. Without this I would have doubts about any significant results.

Thank you for your comments, which are very relevant to our study. Indeed, we had not clearly explained how we intend to use the data from Hansen et al. (2013). We will use the 2000 forest cover data from Hansen et al. (2013), incorporating average tree canopy density thresholds specific to each simplified forest type, as determined by study conducted by the World Resources Institutes (WRI) team in Madagascar, in collaboration with other colleagues in 2023. This study, entitled “Tree Canopy Density for Improved Forest Cover Estimation in Protected Areas of Madagascar” (Rafanoharana et al. 2023), aimed to identify tree canopy density (TCD) thresholds in Madagascar’s protected areas. In addition, the local team works to assess the reliability of Hansen et al. (2013) forest cover data and its updated versions, and ensures that local stakeholders use these data appropriately. Rafanoharana et al.’s paper builds upon the work of Vielledent et al.(2018), who developed a new forest cover map for Madagascar by combining the forest cover data of Harper et al. (2007) with the deforestation data form Hansen et al. (2013). However, in that study, the authors applied the global tree canopy density threshold defined by Hansen et al. (2013) to distinguish forest from non-forest areas. Rafanoharana et al. (2023), in turn, used the updated map from Vieilledent et al. (2018) to develop TCD thresholds specifically adapted to protected areas.

As noted by Rafanoharana et al. (2023), Madagascar hosts a wide diversity of forest types, and applying a standard tree canopy density threshold leads to inaccurate estimates of forest typologies. In their study, Rafanoharana et al. (2023) identified specific tree canopy density thresholds for 111 protected areas listed in the 2015 database of Madagascar’s Protected Areas System. However, they found that 20 of these protected areas consisted of non-contiguous parcels, which required identifying a specific threshold for each parcel.

In our approach, we will use the average tree canopy density thresholds of protected areas belonging to the same simplified forest type, as identified by Rafanoharana et al. (2023), and apply this average to the entire biome.

Moreover, we are aware of the limitations of the Hansen et al. (2013) data, particularly regarding their forest definition and their use of a standard tree canopy density threshold. This is why we rely on the specific thresholds proposed by Rafanoharana et al. (2023), while maintaining the use of the data from Hansen et al. (2013) to benefit from its temporal granularity.

Finally, masking the Hansen et al. (2013) deforestation data using the Harper et al. (2007) forest cover map for the year 2000 is a great idea. We have added your suggestion to the list of robustness checks in the pre-analysis plan in the line 603.

3) I don’t think it’s necessary to run a robustness check using cells of 5km2, I think that’s too large and doesn’t correspond to the scale of land use decision making. Personally, I also would use 1ha as the main scale of analysis. For me, alignment with the scale of land use decision making is more important that alignment with the scale of all covariates. At a scale of 1ha, ~10 neighbouring units will fall within the same 1km2 pixel in the population density and accessibility data. Therefore, all these 10 units will have the same value for these variables. I think that’s okay because in reality, the true values of these variables will differ little within 1km2, especially for the distance variable (which will vary by up to 1km). While I personally would use 1ha as the main scale of analysis and 1km as a robustness check, I will leave it to your discretion which one you use as the main scale. I am happy as long as you do run the analysis at 1 ha resolution (as main or additional).

Indeed, for our study, the choice of analysis unit size was the subject of careful consideration. As noted by Avelino et al. (2016), the scale of explanatory variables and the size of observation units can influence treatment effect estimates and introduce a bias known as the Modifiable Areal Unit Problem (MAUP), especially in conservation. In Madagascar, subsistence agriculture is the primary driver of deforestation, and agricultural plots are generally smaller than 1 ha (MAEP & FAO, 2007). However, most of the explanatory variables we will use have a resolution of 1 km², which underlies our initial choice.

Nevertheless, we will also conduct a robustness test using 1 ha analysis units to assess the impact of this scale effect on our results.

5) I have to admit that I’m confused about your leakage analysis. I am not familiar with the Butts paper but you need to make sure your method is sufficiently clear here so that readers do not need to cross-check with another paper. I’ll try to explain what is causing my confusion to help you with this:

Lines 400-402: ‘These new treatment zones will be matched with remote control zones to obtain the magnitude of spillover and leakage effects.’

This suggests that you will run the leakage analysis separately to the main within-PA analysis, setting units within the buffer ring zones as treated (and therefore excluding units within the PAs), matching to similar controls and running the regressions. This is again suggested in Lines 408 when you refer to buffer zone units as the ‘treatment unit’. And in Lines 433 – 435.

If this is correct you need to make it really clear at the start of the ‘Difference-in-differences with spillover and leakage effects’ section that this is an additional, separate analysis to the main within-PA analysis. It is currently not clear and is confusing. In this additional analysis units within concentric buffer zones outside the PA will be assigned as treated, with treatment occurring in the year the adjacent PA was protected. You can also state in the ‘Treatment Assignment’ section, that you will run two analyses, the first using units from within PAs as treated, and the second using units from a 20km buffer zone around each PA as treated.

Additionally, to make it clear that this is a separate analysis, you should also include the regression equation for the main within-PA analysis in the ‘Basic model of Difference-in-differences’. This will make it clear to readers that the two regressions are different (because the basic model doesn’t include the Ring parameter), therefore this is two separate analyses.

Also, regarding the leakage regression equation shown in Line 411, I’m not convinced about the (1-D) x Ring. If my interpretation is correct, we are interested in the interaction between the ring distance and D = 1, to estimate the effect of a treated buffer zone unit being further from the boundary of a PA (but still within the 20km buffer) after the PA is protected. But 1-1 = 0. So by including (1-D) x Ring aren’t you completely removing the Ring term for the treated units you are interested in?

If I am wrong, and the two analyses are not separate, then you need to explain better how treatment is assigned

We recognize that our description of the spillover and leakage methodology was insufficiently clear. In fact, Butts (2023) does not recommend running a fully separate DiD analysis; rather, spillover and leakage effects are estimated within the same Difference-in-Differences framework. Upon revisiting Butts, we discovered an omission in our regression specification—the coefficient on the ring indicator was missing—and we have corrected that.

To improve clarity, we have made some modifications from line 509 to line 570:

We presented the equation for the Two Way Fixed Effects model in the “Basic model of Difference-in-Differences with Two Way Fixed Effects” in line 522

We added a second equation in the “Difference-in-Differences with Spillover and Leakage Effects” section that explicitly incorporates the ring indicator and its interaction terms in line 557.

We clarified our notation, explaining that D=1 denotes treated units (inside PAs) and therefore (1−D)×Ring captures the interaction for untreated units located in the buffer ring in line 564.

6) In the Heterogeneous Treatment Effects section you state that you will obtain an estimate of the effect of protection within each PA for each year post-intervention. Unless you are planning on conducting an event-study type analysis, which I assume you are not because it is not mentioned anywhere in the text, you will not get an estimate for each year post-protection. Your estimates will represent the average annual change in deforestation in the PA across all years post-protection.

With your study design you will be able to assess spatial heterogeneity, the different performance of different PAs, but not temporal heterogeneity. Including the regression equation for your main analysis would make this very clear.

As such, please delete the text referring to assessing changes in performance over time and instead focus on the patterns, and possible causes of spatial variation in effectiveness.

You are right that we will report the average annual change in deforestation in the PA over the entire post-protection period. However, we also plan to plot the event-study annual coefficients and confidence intervals to assess the plausibility of our findings without interpreting them at this stage. A detailed event-study analysis will be conducted in future works.

7) You need to state the data and code used in the study will be made publicly available (e.g. on Github or FigShare) after the study is complete.

Yes, you are right. We have addressed this in lines 350- 356 stating that the code will be made available on GitHub and on Software Heritage https://www.softwareheritage.org/ , and that the data will be accessible through the DataSuds portal (https://dataverse.ird.fr/).

8) I really like the idea of trying to explain the differential effectiveness of PAs according to a set of hypothesized explanatory factors but I am concerned this might be too much for a single study and may be better reserved for a follow-up.

Thank you for your suggestion. We agree that such analysis would be too much for this study. We plan to have an exploratory analysis that will be a preparation for a more detailed follow-up study.

Detailed comments:

Line 44- 46: The GFW platform uses Hansen et al., (2013) data so directly cite this data source, and delete the reference to Global Forest Watch.

We have rewritten the introduction and reformulate it by mentioning the World Resources Institute in lines 56.

Line 46 – 48: Delete the statement about how Biodiversity can play a crucial role in the developing country as this is too simplistic are currently written. Instead, you can talk about how Madagascar is one of the poorest countries in the world and it’s forests play a crucial role in supporting the livelihoods of its predominantly rural populations by providing ecosystem services.

The correction has been made in lines 63 – 66

Line 4

---

## [Decision Letter · Decision Letter 1]

24 Sep 2025

Dear Dr. Ramiandrisoa,

Thank you for submitting your manuscript to PLOS ONE. After careful consideration, we feel that it has merit but does not fully meet PLOS ONE’s publication criteria as it currently stands. Therefore, we invite you to submit a revised version of the manuscript that addresses the points raised during the review process.

We look forward to receiving your revised manuscript.

Kind regards,

Daniel de Paiva Silva, Ph.D.

Academic Editor

PLOS ONE

Journal Requirements:

Additional Editor Comments:

Dear Dr. Ramiandrisoa,

After this new review round, the final reviewers still raised important issues and hinder the acceptance of your manuscript for publication in PLoS One. Considering this matter, I believe the raised concerns are valid and need further changes. In light of this new decision, I believe you need to go through the issues thoroughly and improve your manuscript accordingly to the improvements needed.

Sincerely,

Daniel Silva

Reviewers' comments:

Reviewer's Responses to Questions

**Comments to the Author**

1. Does the manuscript provide a valid rationale for the proposed study, with clearly identified and justified research questions?

Reviewer #2: Yes

2. Is the protocol technically sound and planned in a manner that will lead to a meaningful outcome and allow testing the stated hypotheses?

Reviewer #2: Partly

3. Is the methodology feasible and described in sufficient detail to allow the work to be replicable?

Reviewer #2: Yes

4. Have the authors described where all data underlying the findings will be made available when the study is complete?

Reviewer #2: Yes

5. Is the manuscript presented in an intelligible fashion and written in standard English?

*PLOS ONE*

Reviewer #2: Yes

You may also provide optional suggestions and comments to authors that they might find helpful in planning their study.

Reviewer #2: Thank you for your courteous response to my comments. The changes you made have significantly improved the clarity of the manuscript. However, I still have a major concern about the DiD models and the assignment of treatment and control that was insufficiently addressed by the revisions. Without properly addressing these comments I cannot recommend this paper be published because I have doubts about the proposed models which have not been adequately explained.

Main point:

I’m sorry but I’m still very unclear on whether the spillover and leakage analysis is a separate analysis to the DiD to estimate the effect of PAs on deforestation inside PAs, shown in Line 400.

To estimate the effect of protection of a Protected Area on deforestation you need to compare outcomes within 1km2 grid cells inside the PA to matched control units which are unaffected by the treatment. To be sure matched control units are unaffected by the treatment they must be outside the 20km buffer zone, to ensure outcomes in these units are not biased by spillover/leakage effects. In this case your treated units come from inside the PA and the control units come from areas outside the 20km buffer zone.

But that means the leakage and spillover analysis and equation in Line 430 must be a separate analysis because you want to look at outcomes within this 20km buffer zone. So, given that, what do the ‘treatment’ and ‘control’ pixels in the leakage and spillover analysis represent?

Lines 451-453 suggest that the leakage and spillover analysis is a separate analysis comparing units within the buffer zone (assigned as ‘treated’) to more distant control units, I’m assuming outside the buffer zone. But if this is true then the notation (1 – D) x Ring is incorrect as the units you are interested in (the buffer units) will be assigned a 1 and that coefficient will be differenced out (because 1-1 = 0).

However, this directly contrasts with your response to my question in the last review where you said:

‘We clarified our notation, explaining that D=1 denotes treated units (inside PAs) and therefore (1−D)×Ring captures the interaction for untreated units located in the buffer ring in line 564.’

If this is correct, then you will be comparing units within protected areas (treatment = 1) to control units within the buffer zone (treatment = 0) to estimate the effect of the PA on deforestation. This is wrong because deforestation in those buffer units can be biased through spillover and leakage effects. Unless, the model controls for that?

This is the most important part of your analysis and it is currently not well enough explained such that it raises some red flags (concerns). Until this is clarified I cannot recommend this article is published.

I hope my points make sense. If not, I would urge you to ask colleagues to review the paper and see what they think. They may be able to explain my concerns.

Specific points

• Line 65: Delete ‘the country’

• Line72-73: I thought orphan parks were the other way around, they had legal status as a PA but no management authority? Please check and revise if I am correct?

• Lines 78- 80: Sorry, I still don’t like this sentence. It would be much better to say that X% show a negative effect, which means that they reduce deforestation, however, Y% were shown to have no effective while Z% conversely increased deforestation.

• Line 95: To clarify consider rephrasing to: ‘… Desbureaux et al showed how PAs displaced deforestation into the surrounding landscape, partially offsetting the observed reductions in deforestation inside the PAs.’

• Move the explanation of matching in Lines 123-124 to Lines 116. This is a broader explanation which should be followed by a more specific explanation of the exact matching algorithm you use (Coarsened Exact matching).

• Line 166: Add ‘which influence the probability of receiving the treatment (i.e. protection) and outcomes, i.e. deforestation.

• Line 192: Change ‘higher’ to ‘lower’ resolution. While the pixel area is higher, this corresponds to lower resolution.

• Line 226 – 229: More recent versions of the accessibility data by Nelson et al., 2019 (https://www.nature.com/articles/s41597-019-0265-5) provide travel time estimates for different population size classes, providing data on distance to cities of 5,000, 15,000, 50,000 people and higher. As accessibility here is a measure of distance to markets and demand, I recommend using this updated dataset and distance to cities with >5,000 people. This is because towns and cities with population size 5,000 – 50,000 will still exert considerable demand pressures on forest land and resources.

• Lines 234 – 243: Please describe what you will do in cases of spatially overlapping PAs? For example, I can see on the WDPA data (protectedplanet.net) that Tsaratanana National Park overlaps with the Complexe des AP Ambohimirahavavy Marivorahona.

• Line 272: Define what the acronym TCD is

• Line 278: State and reference where you are getting these simplified forest types from

• Lines 279 – 280: I recommend saying ‘ we will then apply the specific threshold for each forest type to the Hansen et al., tree canopy cover map of 2000 to create a map of baseline forest cover in 2000 tailored to each forest type. You will analyse deforestation within this 2000 forest extent.’

• Line 286: What polygon units will you use to define non protected areas. Protected Areas have a defined boundary but not protected areas do not.

• Line 299: Add ‘and also influence deforestation’.

• Line 319: Why? Add something about these lands being suitable or marginal for productive uses because they are high elevation, far from markets, or have poor quality soils. Cite Joppa, Lucas N., and Alexander Pfaff. "High and far: biases in the location of protected areas." PloS one 4, no. 12 (2009): e8273.

• Line 332-334: Delete this because I don’t think it is correct. The control variables do not ensure similarity between treatment and control groups – this is the role of the matching variables.

• Control Variables – are these calculated at annual resolution? Please state the time scale at which these variables are measured and included in the model

• Lines 371 – 375: 30% is a high ‘acceptable’ threshold for loss of treated units. Losing treated units which cannot be matched means that your data no longer represents a population but a sample of the population. Un-matched treated units may be non-randomly distributed (e.g. they are all the highest elevation parts of a PA) which can bias the ATT if you exclude these areas. You are correct that there are trade-offs between balance and finding matches for the majority of treated units. However, I think 30% is too high a threshold for acceptable loss of treated units and would strongly recommend reducing it to 10%. You can increase it to say 20% if absolutely necessary if retaining 90% of treated observations does not produce a SMD of less than 0.2.

• Line 392: ‘evolve’ is not the right word here. Try ‘show the same trend in outcomes before the intervention, meaning it is reasonable to assume that they would have continued to follow the same trend in the absence of the intervention’.

• Line 463: This robustness test is not to measure differences caused by spatial autocorrelation but to test the effect of the choice of scale. Please change this sentence to state this.

**Do you want your identity to be public for this peer review?** For information about this choice, including consent withdrawal, please see our Privacy Policy

Reviewer #2: No

---

## [Author Response · Author response to Decision Letter 2]

10 Nov 2025

Thank you for your courteous response to my comments. The changes you made have significantly improved the clarity of the manuscript. However, I still have a major concern about the DiD models and the assignment of treatment and control that was insufficiently addressed by the revisions. Without properly addressing these comments I cannot recommend this paper be published because I have doubts about the proposed models which have not been adequately explained.

Thank you very much for your response and for your valuable comments, which helped us further clarify our approach and refine several key aspects of the pre-analysis plan. Regarding the methodology, we have expanded this section to remove any ambiguity and to enhance both the clarity and reproducibility of the study.

Main point:

I’m sorry but I’m still very unclear on whether the spillover and leakage analysis is a separate analysis to the DiD to estimate the effect of PAs on deforestation inside PAs, shown in Line 400.

To estimate the effect of protection of a Protected Area on deforestation you need to compare outcomes within 1km2 grid cells inside the PA to matched control units which are unaffected by the treatment. To be sure matched control units are unaffected by the treatment they must be outside the 20km buffer zone, to ensure outcomes in these units are not biased by spillover/leakage effects. In this case your treated units come from inside the PA and the control units come from areas outside the 20km buffer zone.

But that means the leakage and spillover analysis and equation in Line 430 must be a separate analysis because you want to look at outcomes within this 20km buffer zone. So, given that, what do the ‘treatment’ and ‘control’ pixels in the leakage and spillover analysis represent?

Lines 451-453 suggest that the leakage and spillover analysis is a separate analysis comparing units within the buffer zone (assigned as ‘treated’) to more distant control units, I’m assuming outside the buffer zone. But if this is true then the notation (1 – D) x Ring is incorrect as the units you are interested in (the buffer units) will be assigned a 1 and that coefficient will be differenced out (because 1-1 = 0).

However, this directly contrasts with your response to my question in the last review where you said:

‘We clarified our notation, explaining that D=1 denotes treated units (inside PAs) and therefore (1−D)×Ring captures the interaction for untreated units located in the buffer ring in line 564.’

If this is correct, then you will be comparing units within protected areas (treatment = 1) to control units within the buffer zone (treatment = 0) to estimate the effect of the PA on deforestation. This is wrong because deforestation in those buffer units can be biased through spillover and leakage effects. Unless, the model controls for that?

This is the most important part of your analysis and it is currently not well enough explained such that it raises some red flags (concerns). Until this is clarified I cannot recommend this article is published.

I hope my points make sense. If not, I would urge you to ask colleagues to review the paper and see what they think. They may be able to explain my concerns.

Thank you very much for this thoughtful comment. We understand that the distinction between the two equations may have caused some confusion, and we have clarified this point in the revised version of the manuscript.

The first equation (line 406) corresponds to the standard DiD model, which estimates the effect of protection within PAs. In this specification, treated units are 1km² cells located inside PAs, with control units located more than 20 km away from PA boundaries, to ensure that they are not affected by spillover or leakage effects.

The second equation in line 468, extends this canonical model following Butts (2023) to account for spatial spillovers:

It jointly estimates:

(i) the treatment effect () on treated units (inside PAs); and

(ii) (ii) the spillover effects () on non-treated units located within successive distance rings around PAs (with j = 1 for 0-5km, j = 2 for 5-10km, and j = 3 for 10-20km). Each measures the average change in deforestation within ring j relative to control units located beyond 20 km.

For treated units (), the spillover term is zero (), yielding the estimated effect of protection. For non-treated units located in the buffer, the treatment term is 0 (), and the ring indicator is one (), yielding , the protection spillovers or leakagefor ring j. Both treated and buffer units are compared to a common control group: the cells located beyond 20 km from any PA.

In their review of the recent advances in the state of the art for difference-in-difference methods, Roth et al. (2023) emphasize that standard DiD models rely on the Stable Unit Treatment Value Assumption (SUTVA), which is violated in the presence of spillovers. In the case of spatial spillovers, Roth et al. (2023) point to the Butts approach that we are proposing to adopt, in order to factor in these spillovers in the total effect estimation.

We have clarified this point in the manuscript and now state explicitly that the treatment and spillover effects will be reported separately besides the total effect. The concrete implementation in R will follow the procedure illustrated in the Tennessee Valley Authority replication example in Butts (2023), available at https://github.com/kylebutts/Spatial-Spillover/blob/master/code/TVA/analysis.R

Specific points:

• Line 65: Delete ‘the country’

It has been deleted in line 64.

• Line72-73: I thought orphan parks were the other way around, they had legal status as a PA but no management authority? Please check and revise if I am correct?

Thank you for your comment. You are right, paper parks refer to protected areas without effective management, while orphan parks are those lacking managers. This clarification has been incorporated in lines 70-72.

• Lines 78- 80: Sorry, I still don’t like this sentence. It would be much better to say that X% show a negative effect, which means that they reduce deforestation, however, Y% were shown to have no effective while Z% conversely increased deforestation.

Thank you very much for your comment. We tried to identify the percentage; however, this information is not provided in the original paper. Nevertheless, we have reformulated the sentence to: ‘‘The meta-analysis conducted by Börner et al.(2020) shows that while protected areas can reduce deforestation, they may also have no effect or even increase deforestation in some cases’’ to make it clearer in lines 76-78.

• Line 95: To clarify consider rephrasing to: ‘… Desbureaux et al showed how PAs displaced deforestation into the surrounding landscape, partially offsetting the observed reductions in deforestation inside the PAs.’

It has been rephrased in lines 92-94.

• Move the explanation of matching in Lines 123-124 to Lines 116. This is a broader explanation which should be followed by a more specific explanation of the exact matching algorithm you use (Coarsened Exact matching).

It has been moved in lines 114-116.

• Line 166: Add ‘which influence the probability of receiving the treatment (i.e. protection) and outcomes, i.e. deforestation.

It has been added in lines 164- 166.

• Line 192: Change ‘higher’ to ‘lower’ resolution. While the pixel area is higher, this corresponds to lower resolution.

It has been changed in line 191.

• Line 226 – 229: More recent versions of the accessibility data by Nelson et al., 2019 (https://www.nature.com/articles/s41597-019-0265-5) provide travel time estimates for different population size classes, providing data on distance to cities of 5,000, 15,000, 50,000 people and higher. As accessibility here is a measure of distance to markets and demand, I recommend using this updated dataset and distance to cities with >5,000 people. This is because towns and cities with population size 5,000 – 50,000 will still exert considerable demand pressures on forest land and resources.

Thank you very much for your suggestion. We agree with the advantages of the Nelson et al. (2019) dataset and it is the one that we initially planned to use. However, we realized that this dataset was built with 2015 as reference year, which poses an endogeneity problem and led us to favor the Uchida and Nelson (2011) dataset, as it used 2000 as reference year. Our analysis focuses on protected areas established between 2002 and 2022 and according to Ho et al. (2007), the covariates must either not be plausibly affected by the treatment or be measured prior to treatment. The construction or maintenance of roads, are likely affected by the presence of PAs (depending on the settings, to comply with conservation rules and avoid anthropic pressure, or to facilitate tourism), therefore we need accessibility data that predates our study period.

• Lines 234 – 243: Please describe what you will do in cases of spatially overlapping PAs? For example, I can see on the WDPA data (protectedplanet.net) that Tsaratanana National Park overlaps with the Complexe des AP Ambohimirahavavy Marivorahona.

If two PAs overlap, or if a recent PA is established within the buffer of an older one, priority is given to the older protected area in order to ensure that there is only one treatment. This clarification has been added in lines 235-238.

• Line 272: Define what the acronym TCD is

The acronym TCD means Tree Canopy Density. It was alread defined on line 130 and we added another definition in line 272.

• Line 278: State and reference where you are getting these simplified forest types from

The simplified forest types used in our study are derived from the TEOW datasets. It has been stated in line 279

• Lines 279 – 280: I recommend saying ‘we will then apply the specific threshold for each forest type to the Hansen et al., tree canopy cover map of 2000 to create a map of baseline forest cover in 2000 tailored to each forest type. You will analyse deforestation within this 2000 forest extent.’

It has been rephrased in lines 280-282.

• Line 286: What polygon units will you use to define non protected areas. Protected Areas have a defined boundary but not protected areas do not.

The polygon units representing unprotected areas correspond to 1km² grid cells located outside the boundaries of protected areas. This clarification has been added in lines 288-289.

• Line 299: Add ‘and also influence deforestation’.

It has been added in line 302.

• Line 319: Why? Add something about these lands being suitable or marginal for productive uses because they are high elevation, far from markets, or have poor quality soils. Cite Joppa, Lucas N., and Alexander Pfaff. "High and far: biases in the location of protected areas." PloS one 4, no. 12 (2009): e8273.

Thank you for your suggestion. We already cite this Joppa and Pfaff in this paragraph (it is the reference 43). As this paragraph focuses on slope and altitude, we added an explanation on lines 322-325 to specify that we refer here to the opportunity costs for resource extraction and agricultural activities. We also cite this same reference 43 (Joppa and Pfaff, 2009) in 3 other instances, among them the paragraphs referring to accessibility and population density variables.

• Line 332-334: Delete this because I don’t think it is correct. The control variables do not ensure similarity between treatment and control groups – this is the role of the matching variables.

It has been deleted.

• Control Variables – are these calculated at annual resolution? Please state the time scale at which these variables are measured and included in the model

The control variables are calculated at an annual resolution based on monthly data. The drought indicator is available from 1997 to 2023 and is computed annually with a three-year lag to capture delayed ecological responses. The cyclone exposure variable is available from 1999 to 2023 and is computed annually with a one-year lag to account for the consequences of previous cyclones. These are specified in lines 337, 347-348, 351- 353.

• Lines 371 – 375: 30% is a high ‘acceptable’ threshold for loss of treated units. Losing treated units which cannot be matched means that your data no longer represents a population but a sample of the population. Un-matched treated units may be non-randomly distributed (e.g. they are all the highest elevation parts of a PA) which can bias the ATT if you exclude these areas. You are correct that there are trade-offs between balance and finding matches for the majority of treated units. However, I think 30% is too high a threshold for acceptable loss of treated units and would strongly recommend reducing it to 10%. You can increase it to say 20% if absolutely necessary if retaining 90% of treated observations does not produce a SMD of less than 0.2.

Thank you very much for your constructive recommendation. We fully agree that excluding unmatched treated units may introduce bias in the estimated ATT if those units are not randomly distributed. Your reply led us to substantially revise the proposed matching procedure.

In the revised version, we have therefore tightened the maximum acceptable exclusion threshold from 30 % to 20 %, while specifying that the optimal specification that reaches a SMD ≤ 0.25 while retaining the largest possible share of treated units. In addition, for any protected area where achieving satisfactory balance requires discarding more than 10 % of treated units, we will also report an alternative estimate based on a coarser binning (SMD > 0.25) that retains at least 90 % of treated units.

We have also revised the target SMD, which was initially set at 0.2, and adopted 0.25 as an acceptable threshold, following the criterion used by Devenish et al. (2022, p. 505). Although the difference is modest, this adjustment should reduce the number of excluded treated units while remaining within the range generally considered indicative of satisfactory covariate balance.

We have also removed the requirement that all continuous variables share the same number of bins. We think that this modification will avoid unnecessary data fragmentation and better accommodates differences in the empirical distributions of covariates, such as between skewed variables (e.g., accessibility, population density) and more normally distributed ones (e.g., slope, elevation).

These revisions have been added in lines 370–394 of the revised manuscript.

• Line 392: ‘evolve’ is not the right word here. Try ‘show the same trend in outcomes before the intervention, meaning it is reasonable to assume that they would have continued to follow the same trend in the absence of the intervention’.

Thank you for your recommendation. This statement has been incorporated in line 400-402.

• Line 463: This robustness test is not to measure differences caused by spatial autocorrelation but to test the effect of the choice of scale. Please change this sentence to state this.

Thank you for your comment, it has been changed in lines 508-509.

---

## [Decision Letter · Decision Letter 2]

22 Dec 2025

Dear Dr.  Ramiandrisoa,

Thank you for submitting your manuscript to PLOS ONE. After careful consideration, we feel that it has merit but does not fully meet PLOS ONE’s publication criteria as it currently stands. Therefore, we invite you to submit a revised version of the manuscript that addresses the points raised during the review process.

We look forward to receiving your revised manuscript.

Kind regards,

Daniel de Paiva Silva, Ph.D.

Academic Editor

PLOS One

Journal Requirements:

Additional Editor Comments:

Dear Dr. Ramiandrisoa,

After this new review round, you will see that the reviewer accepted your manuscript for publication pending minor reviews. Therefore, please proceed with the minor changes suggested by the reviewers and consider your manuscript accepted after you do this.

Congratulations on your hard work.

Happy Holidays!

Daniel Silva

Reviewer's Responses to Questions

**Comments to the Author**

1. Does the manuscript provide a valid rationale for the proposed study, with clearly identified and justified research questions?

Reviewer #2: Yes

2. Is the protocol technically sound and planned in a manner that will lead to a meaningful outcome and allow testing the stated hypotheses?

Reviewer #2: Yes

3. Is the methodology feasible and described in sufficient detail to allow the work to be replicable?

Reviewer #2: Yes

4. Have the authors described where all data underlying the findings will be made available when the study is complete?

Reviewer #2: Yes

5. Is the manuscript presented in an intelligible fashion and written in standard English?

*PLOS ONE*

Reviewer #2: Yes

You may also provide optional suggestions and comments to authors that they might find helpful in planning their study.

Reviewer #2: Thank you. The changes you have made to the text in response my previous comments have made the analytical strategy a lot clearer and easier to understand. And I appreciate the decision algorithm you have included in the matching process to try and minimise the loss of treated units while maintaining balance. That’s a really good idea.

I can now recommend that this article is accepted, pending clarification of how percentage deforestation will be measured (see comment about Line 267-268). This is the only essential change.

I have a few additional small comments to improve the clarity of the text but these are very minor changes which should not take more than a few minutes to address.

Line 63: Change ‘beyond’ to ‘behind’

Line 91: Is it a reduction in deforestation ‘risk’ (which implies probability) or the estimated reduction in absolute deforestation?

Line 100-101: Please clarify whether this decrease was relative to the previous period, or relative to a control/counterfactual, or both?

Line 116 – 118: is it for each variable that researchers fix ex-ante the imbalance between treated and control? If so, please add this to this sentence?

Line 175: The matched controls won’t have exactly the same observable characteristics so please change to say something like ‘which are similar as possible in terms of observable characteristics (…) to the PAs’.

Line 192 – 193: Forest cover, slope and elevation are measured at 0.09 ha, not 1ha. 30x30 = 900m2 = 0.09 ha

Line 237 – 239: Does this mean that the year of treatment is assigned as the year the older park was created?

Line 267-268: Will deforestation be measured as a percentage of grid cell area or baseline forest cover?

Line 287: Change to ‘In this study, the treated units are 1km grid cells within PAs and the control units are grid cells in unprotected areas outside the boundaries of any protected area.’

Line 328: Change ‘area’ to ‘1km grid cell’

Line 362: Are units assigned to bins based on the coarsened values of all of their covariates? If so, add ‘all’ to this sentence

Line 364 – 365: You can delete this sentence because it is repetitive

Line 371: Add ‘between treatment and matched controls’ after ‘Standardised Mean Difference’.

Line 430: Add the qualifier ‘potential’ before ‘occurrence of spillover and leakage effects’

Line 435: Add ‘from the PA’ at the end of this sentence.

Line 436: Change ‘the total effect on the treated units’ to ‘the total effect of protection on treated units within the PA’.

**Do you want your identity to be public for this peer review?** For information about this choice, including consent withdrawal, please see our Privacy Policy

Reviewer #2: No

---

## [Author Response · Author response to Decision Letter 3]

14 Jan 2026

RESPONSE TO EDITOR AND REVIEWERS

Dear Dr Daniel Silva,

Thank you for the opportunity to submit a new revised version of our manuscript entitled: “Impact of Protected Areas on Deforestation in Madagascar from 2000 to 2023: A Pre-Analysis Plan.”

We greatly appreciate the time and effort that you and the reviewers have devoted to evaluating our work since the first submission. We are grateful to the reviewers for their insightful comments and for the decision of minor revisions.

We have incorporated the changes in accordance with the suggestions and comment provided.

Below, we present a point-by-point response to the reviewers’ comments and recommendations.

Response to reviewers:

Reviewer #2:

Thank you. The changes you have made to the text in response my previous comments have made the analytical strategy a lot clearer and easier to understand. And I appreciate the decision algorithm you have included in the matching process to try and minimise the loss of treated units while maintaining balance. That’s a really good idea.

I can now recommend that this article is accepted, pending clarification of how percentage deforestation will be measured (see comment about Line 267-268). This is the only essential change.

I have a few additional small comments to improve the clarity of the text but these are very minor changes which should not take more than a few minutes to address.

Thank you very much for your message and for the valuable comments and suggestions. They have been extremely helpful in improving the clarity of our study and strenghtening our analytical approach.

We also sincerely appreciate your recommendation for acceptance, it is very encouraging.

In the revised manuscript, we have carefully addressed your comments, as detailed below :

Line 63: Change ‘beyond’ to ‘behind’

It has been changed in line 62.

Line 91: Is it a reduction in deforestation ‘risk’ (which implies probability) or the estimated reduction in absolute deforestation?

Thank you for your comment. Indeed, in the Discussion and Conclusion sections, Desbureaux et al. [1] refer to a 20 % reduction in deforestation, not to deforestation risk. We therefore removed the word “risk” at line 91.

Line 100-101: Please clarify whether this decrease was relative to the previous period, or relative to a control/counterfactual, or both?

Thank you for highlighting that our phrasing was ambiguous. The result reported by Eklund et al. [2] stems from the comparison of two distinct cross-sectional analysis for the subsequent decades. They compared 10-years deforestation rates between matched controls and treatment pixels on the one hand for the period 1990-2000 and on the other hand for the period 2000-2010. They do not use the diachronic feature of their dataset to implement a Difference-in-Difference approach or a similar approach. What they do is observe the variation between both decade of overall estimated impact for each biome. Our phrasing" This method, like counterfactual approaches,...", could imply that their method was not counterfactual", so we replaced it by "this counterfactual method". We also rephrased the sentence "More precisely, they found that deforestation decreased from 1990-2000 to 2000-2010 in humid forest but increased in dry and spiny forests" to describe more precisely their analytical in lines 100-104.

Line 116 – 118: is it for each variable that researchers fix ex-ante the imbalance between treated and control? If so, please add this to this sentence?

Indeed, it is for each variable that researchers fix ex-ante the imbalance between treated and control according to Iacus et al.[3] in 2009 and Iacus et al. [4] in 2012. This clarification has been made by adding « for each variable » in line 121.

Line 175: The matched controls won’t have exactly the same observable characteristics so please change to say something like ‘which are similar as possible in terms of observable characteristics (…) to the PAs’.

You are right. The proposed sentence has been added in lines 177-178.

Line 192 – 193: Forest cover, slope and elevation are measured at 0.09 ha, not 1ha. 30x30 = 900m2 = 0.09 ha

Thank you very much. We have corrected the resolution to 0.09 ha in line 196.

Line 237 – 239: Does this mean that the year of treatment is assigned as the year the older park was created?

Yes, exactly. We have made it clearer by rephasing the sentence to “ In such cases, the treatment year is assigned as the year the older park is created to ensure that there is only one treatment” in lines 240-242.

Line 267-268: Will deforestation be measured as a percentage of grid cell area or baseline forest cover?

Thank you very much for your question. Indeed the method used to compute the outcome was not yet sufficiently clear. We will download the forest cover data using the mapme.biodiversity package, which derives these measures from Global Forest Watch data based on Hansen et al.[5] . In this package, forest cover is expressed as the percentage of each grid’s area, on a 0-100 scale (0 = no forest; 100 = completely forested).

In our study, we will start from the percentage of each grid cell covered by forest in 2000. The annual percentage of deforestation will then be computed, for each grid cell, as the share of the its area lost between year n and n+1.

For clarity, we added the following sentence in lines 274-276: “The annual percentage of deforestation will be computed for each grid cell, as the share of its forested area at year n that is lost by year n+1”.

We also replaced the word “intact” with “forested” in line 282 to ensure consistency with the Tree Canopy Density threshold approach.

Line 287: Change to ‘In this study, the treated units are 1km grid cells within PAs and the control units are grid cells in unprotected areas outside the boundaries of any protected area.’

It has been changed in lines 293-295.

Line 328: Change ‘area’ to ‘1km grid cell’

It has been changed to 1 km² grid cell in lines 332-333.

Line 362: Are units assigned to bins based on the coarsened values of all of their covariates? If so, add ‘all’ to this sentence

Yes, the units are assigned to bins based on the coarsened values of all their covariates. The word « all » has been added to the sentence in line 367.

Line 364 – 365: You can delete this sentence because it is repetitive

It has been deleted.

Line 371: Add ‘between treatment and matched controls’ after ‘Standardised Mean Difference’.

It has been added in 375.

Line 430: Add the qualifier ‘potential’ before ‘occurrence of spillover and leakage effects’.

It has been added in line 435.

Line 435: Add ‘from the PA’ at the end of this sentence.

It has been added in line 440.

Line 436: Change ‘the total effect on the treated units’ to ‘the total effect of protection on treated units within the PA’.

It has been changed in lines 441-442.

Referecences cited in the response

1. Desbureaux S, Aubert S, Brimont L, Karsenty A, Lohanivo AC, Rakotondrabe M, et al. The impact of Protected Areas on Deforestation? An Exploration of the Economic and Political Channels for Madagascar’s Rainforests (2001-12). Working Papers. HAL; 2015 July. Report No.: hal-01176860. Available: https://ideas.repec.org/p/hal/wpaper/hal-01176860.html

2. Eklund J, Blanchet FG, Nyman J, Rocha R, Virtanen T, Cabeza M. Contrasting spatial and temporal trends of protected area effectiveness in mitigating deforestation in Madagascar. Biological Conservation. 2016;203: 290–297.

3. Iacus SM, King G, Porro G. cem : Software for Coarsened Exact Matching. J Stat Soft. 2009;30. doi:10.18637/jss.v030.i09

4. Iacus SM, King G, Porro G. Causal inference without balance checking: Coarsened exact matching. Political analysis. 2012;20: 1–24.

5. Hansen MC, Potapov PV, Moore R, Hancher M, Turubanova SA, Tyukavina A, et al. High-resolution global maps of 21st-century forest cover change. Science. 2013;342: 850–853. Following the revisions made to the manuscript, we provide below the list of the references that were removed and those that were added compared with the first version. These updates were implemented in response to the reviewers’suggestions to improve the internal consistency of the study and strengthen the scientific rigor of the manuscript.

References removed compared with the first version :

• Cardinale BJ, Duffy JE, Gonzalez A, Hooper DU, Perrings C, Venail P, et al. Biodiversity loss and its impact on humanity. Nature. 2012;486: 59–67. doi:10.1038/nature11148

• Méral P, Froger G, Andriamahefazafy F, Rabearisoa A. Chapitre 5. Le financement des aires protégées à Madagascar : de nouvelles modalités. In: Aubertin C, Rodary E, editors. Aires protégées, espaces durables ? Marseille: IRD Éditions; 2013. pp. 135–155. doi:10.4000/books.irdeditions.5677

• Ribas LG dos S, Pressey RL, Loyola R, Bini LM. A global comparative analysis of impact evaluation methods in estimating the effectiveness of protected areas. Biological Conservation. 2020;246: 108595. doi:10.1016/j.biocon.2020.108595

• Hansen MC, Loveland TR. A review of large area monitoring of land cover change using Landsat data. Remote Sensing of Environment. 2012;122: 66–74. doi:10.1016/j.rse.2011.08.024

• Fritz S, Laso Bayas JC, See L, Schepaschenko D, Hofhansl F, Jung M, et al. A Continental Assessment of the Drivers of Tropical Deforestation With a Focus on Protected Areas. Frontiers in Conservation Science. 2022;3. Available: https://www.frontiersin.org/article/10.3389/fcosc.2022.830248

• Danko D M. Proceedings of the Eleventh Annual ESRI User Conference. Environmental Systems Research Institute; 1991.

• Knapp KR, Kruk MC, Levinson DH, Diamond HJ, Neumann CJ. The International Best Track Archive for Climate Stewardship (IBTrACS): Unifying Tropical Cyclone Data. Bulletin of the American Meteorological Society. 2010;91: 363–376. doi:10.1175/2009BAMS2755.1

• Gahtan J, Knapp KR, Schreck CJI, Diamond HJ, Kossin JP, Kruk MC. International Best Track Archive for Climate Stewardship (IBTrACS) Project, Version 4.01. 2024 [cited 21 Nov 2024]. Available: https://www.ncei.noaa.gov/access/metadata/landing-page/bin/iso?id=gov.noaa.ncdc:C01552

• Eklund J, Jones JP, Räsänen M, Geldmann J, Jokinen A-P, Pellegrini A, et al. Elevated fires during COVID-19 lockdown and the vulnerability of protected areas. Nature Sustainability. 2022;5: 603–609.

• Andrianambinina FOD, Waeber PO, Schuurman D, Lowry PP, Wilmé L. Clarification on protected area management efforts in Madagascar during periods of heightened uncertainty and instability. Madagascar Conservation & Development. 2022;17: 25–28.

• Goodman SM, Raherilalao MJ, Wohlhauser S, Rabenandrasana JCN, Rakotondratsimba HM, Andriamialisoa F, et al. Les aires protégées terrestres de Madagascar: leur histoire, description et biote. Association Vahatra; 2018. Available: https://zoboko.com/publisher/association-vahatra-in-antananarivo

• R Core Team. R: A Language and Environment for Statistical Computing. Vienna, Austria: R Foundation for Statistical Computing; 2021. Available: https://www.R-project.org/

• Jones JPG, Mandimbiniaina R, Kelly R, Ranjatson P, Rakotojoelina B, Schreckenberg K, et al. Human migration to the forest frontier: Implications for land use change and conservation management. Geo: Geography and Environment. 2018;5: e00050. doi:10.1002/geo2.50

• Andam KS, Ferraro PJ, Pfaff A, Sanchez-Azofeifa GA, Robalino JA. Measuring the effectiveness of protected area networks in reducing deforestation. Proceedings of the National Academy of Sciences. 2008;105: 16089–16094. doi:10.1073/pnas.0800437105

• Saha S, Saha M, Mukherjee K, Arabameri A, Ngo PTT, Paul GC. Predicting the deforestation probability using the binary logistic regression, random forest, ensemble rotational forest, REPTree: A case study at the Gumani River Basin, India. Science of The Total Environment. 2020;730: 139197. doi:10.1016/j.scitotenv.2020.139197

• Ferraro PJ, Hanauer MM, Sims KRE. Conditions associated with protected area success in conservation and poverty reduction. Proceedings of the National Academy of Sciences. 2011;108: 13913–13918. doi:10.1073/pnas.1011529108

References added compared with the first version :

• Achard F, DeFries R, Eva H, Hansen M, Mayaux P, Stibig H-J. Pan-tropical monitoring of deforestation. Environ Res Lett. 2007;2: 045022. doi:10.1088/1748-9326/2/4/045022

• Tabor K, Hewson J, Tien H, González-Roglich M, Hole D, Williams JW. Tropical Protected Areas Under Increasing Threats from Climate Change and Deforestation. Land. 2018;7: 90. doi:10.3390/land7030090

• Vieilledent G, Grinand C, Vaudry R. Forecasting deforestation and carbon emissions in tropical developing countries facing demographic expansion: a case study in M adagascar. Ecology and Evolution. 2013;3: 1702–1716. doi:10.1002/ece3.550

• Andam KS, Ferraro PJ, Pfaff A, Sanchez-Azofeifa GA, Robalino JA. Measuring the effectiveness of protected area networks in reducing deforestation. Proceedings of the National Academy of Sciences. 2008;105: 16089–16094. doi:10.1073/pnas.0800437105

• Ma B, Zhang Y, Hou Y, Wen Y. Do Protected Areas Matter? A Systematic Review of the Social and Ecological Impacts of the Establishment of Protected Areas. IJERPH. 2020;17: 7259. doi:10.3390/ijerph17197259

• Myers N. Threatened biotas: “Hot spots” in tropical forests. Environmentalist. 1988;8: 187–208. doi:10.1007/BF02240252

• World Bank. Évaluation de la pauvreté à Madagascar: Naviguer sur deux décennies de pauvreté élevée et tracer la voie du changement. In: World Bank [Internet]. 2024 [cited 8 Oct 2024]. Available: https://www.banquemondiale.org/fr/country/madagascar/publication/madagascar-afe-poverty-assessment-navigating-two-decades-of-high-poverty-and-charting-a-course-for-change

• Gardner CJ, Nicoll ME, Birkinshaw C, Harris A, Lewis RE, Rakotomalala D, et al. The rapid expansion of Madagascar’s protected area system. Biological Conservation. 2018;220: 29–36. doi:10.1016/j.biocon.2018.02.011

• Iacus SM, King G, Porro G. Causal inference without balance checking: Coarsened exact matching. Political analysis. 2012;20: 1–24.

• Iacus SM, King G, Porro G. Multivariate Matching Methods That Are Monotonic Imbalance Bounding. Journal of the American Statistical Association. 2011;106: 345–361. doi:10.1198/jasa.2011.tm09599

• Roth J, Sant’Anna PHC, Bilinski A, Poe J. What’s trending in difference-in-differences? A synthesis of the recent econometrics literature. Journal of Econometrics. 2023;235: 2218–2244. doi:10.1016/j.jeconom.2023.03.008

• Butts K, Gardner J. {did2s}: Two-Stage Difference-in-Differences. The R Journal. 2022;14: 162–173. doi:10.32614/RJ-2022-048

• Callaway B, Sant’Anna PHC. Difference-in-Differences With Multiple Time Periods and an Application on the Minimum Wage and Employment. SSRN Journal. 2018 [cited 15 July 2024]. doi:10.2139/ssrn.3148250

• Jones JPG, Rakotonarivo OS, Razafimanahaka JH. Forest Conservation in Madagascar: Past, Present, and Future. 2021.

• Scales IR. The drivers of deforestation and the complexity of land use in Madagascar. Conservation and environmental management in Madagascar. Routledge; 2014. pp. 129–150.

• Stuart EA. Matching methods for causal inference: A review and a look forward. Stat Sci. 2010;25: 1–21. doi:10.1214/09-STS313

• Joppa L, Pfaff A. Reassessing the forest impacts of protection: The challenge of nonrandom location and a corrective method. Annals of the New York Academy of Sciences. 2010;1185: 135–149. doi:10.1111/j.1749-6632.2009.05162.x

• Angrist JD, Pischke J-S. Mostly Harmless Econometrics: An Empiricist’s Companion. 1st edition. Princeton, New Jersey Oxford: Princeton University Press; 2009.

• Baker A, Callaway B, Cunningham S, Goodman-Bacon A, Sant’Anna PHC. Difference-in-Differences Designs: A Practitioner’s Guide. arXiv; 2025. doi:10.48550/arXiv.2503.13323

• Callaway B, Sant’Anna PHC. Difference-in-Differences with multiple time periods. Journal of Econometrics. 2021;225: 200–230. doi:10.1016/j.jeconom.2020.12.001

• Abadie A. Semiparametric Difference-in-Differences Estimators. The Review of Economic Studies. 2005;72: 1–19. doi:10.1111/0034-6527.00321

• Desbureaux S. Su

---

## [Editor Report · Decision Letter 3]

19 Jan 2026

Impact of protected areas on deforestation in Madagascar from 2000 to 2023: A pre-analysis plan

PONE-D-25-03411R3

Dear Dr. Ramiandrisoa,

We’re pleased to inform you that your manuscript has been judged scientifically suitable for publication and will be formally accepted for publication once it meets all outstanding technical requirements.

Kind regards,

Daniel de Paiva Silva, Ph.D.

Academic Editor

PLOS One

Additional Editor Comments (optional):

Congratulations on the hard work you and your co-authors employed to improve the manuscript!

Happy 2026!

Sincerely,

Daniel Silva
---

## [Editor Report · Acceptance letter]

PONE-D-25-03411R3

PLOS One

Dear Dr. Ramiandrisoa,

I'm pleased to inform you that your manuscript has been deemed suitable for publication in PLOS One. Congratulations! Your manuscript is now being handed over to our production team.

Kind regards,

on behalf of

Dr. Daniel de Paiva Silva

Academic Editor

PLOS One